# Associative memory neurons of encoding multi-modal signals are recruited by neuroligin-3-mediated new synapse formation

Yang Xu, Tian-liang Cui, Jia-yi Li, Bingchen Chen, Jin-Hui Wang*

College of Life Science, University of Chinese Academy of Sciences, Beijing, China

**\*For correspondence:**
wangjinhui@ucas.ac.cn

**Competing interest:** The authors declare that no competing interests exist.

**Abstract** The joint storage and reciprocal retrieval of learnt associated signals are presumably encoded by associative memory cells. In the accumulation and enrichment of memory contents in lifespan, a signal often becomes a core signal associatively shared for other signals. One specific group of associative memory neurons that encode this core signal likely interconnects multiple groups of associative memory neurons that encode these other signals for their joint storage and reciprocal retrieval. We have examined this hypothesis in a mouse model of associative learning by pairing the whisker tactile signal sequentially with the olfactory signal, the gustatory signal, and the tail-heating signal. Mice experienced this associative learning show the whisker fluctuation induced by olfactory, gustatory, and tail-heating signals, or the other way around, that is, memories to multi-modal associated signals featured by their reciprocal retrievals. Barrel cortical neurons in these mice become able to encode olfactory, gustatory, and tail-heating signals alongside the whisker signal. Barrel cortical neurons interconnect piriform, S1-Tr, and gustatory cortical neurons. With the barrel cortex as the hub, the indirect activation occurs among piriform, gustatory, and S1-Tr cortices for the second-order associative memory. These associative memory neurons recruited to encode multi-modal signals in the barrel cortex for associative memory are downregulated by *neuroligin-3* knock-down. Thus, associative memory neurons can be recruited as the core cellular substrate to memorize multiple associated signals for the first-order and the second-order of associative memories by neuroligin-3-mediated synapse formation, which constitutes neuronal substrates of cognitive activities in the field of memoriology.

## eLife assessment

Multimodal experiences that for example contain both visual and tactile components are encoded as associative memories. This manuscript is a **valuable** contribution supporting structural and functional brain plasticity following associative training protocols that pair together different types of sensory stimuli. The results provide **solid** support for this plasticity being a basis for cross-modal associative memories.

## Introduction

Associative learning and memory are major forms of the information acquisition and storage (***Byrne, 1987***; ***Wasserman and Miller, 1997***), in which multiple cross-modal signals are learnt associatively and memorized jointly (***Feng et al., 2017***; ***Vincis and Fontanini, 2016***; ***Wang et al., 2015***; ***Wang et al., 2013***; ***Wang et al., 2019b***). Memories to associated signals have been thought to be essential for the cognitive activities and emotional responses (***Wang et al., 2019b***; ***Kandel and Pittenger, 1999***;

*Poo et al., 2016*; *Silva et al., 2009*). The cellular mechanism underlying associative memory was presumably based on an activity-dependent strengthening in the interconnection of cell assemblies (*Hebb, 1949*), which has been supported by synaptic and neuronal plasticity during the learning and memory (*Armano et al., 2000*; *Bliss and Lømo, 1973*; *Bliss and Lynch, 1988*; *Lisman et al., 2018*; *Wang et al., 1997*; *Zhang et al., 2004*). Although the neuronal plasticity influences the strengths in the formation of associative memory and the retrieval of memorized signals, this activity-dependent plasticity in a single neural pathway cannot interpret the associative memory featured by the joint storage and the reciprocal retrieval of associated signals specifically inputted from multiple pathways (*Wang et al., 2019b*). Recent studies have discovered the recruitment of associative memory neurons in mice while they memorize the associatively learnt signals. These associative memory neurons show the synapse interconnections morphologically among coactive cross-modal cerebral cortices and encode the synapse signals functionally received from these cortices (*Feng et al., 2017*; *Wang et al., 2015*; *Gao et al., 2019*; *Liu et al., 2017*; *Yan et al., 2016*; *Gao et al., 2016* ; ). These data have been supported by causal relationships between the recruitment of associative memory neuron and the formation of associative memory (*Feng et al., 2017*; *Lei et al., 2017*; *Wu et al., 2020*). Whether these associative memory neurons can be recruited with more extensive cross-modal interconnections and work as the core to translate the multiple signals directly and indirectly for the first order and second order of associative memory remains tested to strengthen the principle of coactivity together and interconnections together (*Wang et al., 2019b*; *Wang et al., 2018*; *Wang et al., 2019b*).

In the accumulation of memory contents during the postnatal growth, a signal may become the core signal shared by other signals for their reciprocal retrieval (*Wang et al., 2019b*; *Schacter et al., 2020*). The auditory signal of 'apple' is shared with the visual signal about its shape/color, the gustatory signal about its taste, and the olfactory signal about its smell. The listening of this 'apple' sound by the auditory sensory modality can induce the reciprocal recalls of other apple's features in different modalities. The thunderstorm signal often associates with other signals, for example, a heavy rain, something wetness, and flooding. A thunderstorm can lead to the logical reasoning about the forthcoming of the heavy rain and flooding in the cognition as well as the worry about their happening in the emotion. In terms of cellular bases of these memory retrievals, cognition, and emotion, we assume that neuronal assemblies in a single modality cortex may interconnect neuronal assemblies in other modality cortices by their coactivation (*Feng et al., 2017*; *Wang et al., 2019b*) and that these neuronal assemblies become able to encode all of these associated signals (*Feng et al., 2017*; *Vincis and Fontanini, 2016*; *Wang et al., 2015*), that is, the recruitment of the associative memory neurons to encode multi-modal signals in the associative learning during the postnatal growth (*Wang et al., 2019b*). By the repetitive activations of these connected neurons in the retrievals of the associated signals, the interconnections of associative memory neurons in a single modality cortex with those associative memory neurons in other modality cortices may be further strengthened (*Hebb, 1949*). Through the activity-dependent interconnection and strengthening, associative memory neurons in this single modality may become a core station to translate other signals by jumping over the shared signal, or the retrieval of associative memory in the cross-core manner indirectly. For instance, the apple's shape signal in the mind retrieves its taste signal or odor signal with jumping over the apple's sound signal. The heavy rain signal may retrieve the flooding signal without thinking about the thunderstorm signal in the mind. The test of this assumption is critical to reveal the cellular substrate correlated to a wide range of brain functions in the cognition and the emotion in the field of memoriology.

In the present study, we intend to investigate the recruitment of associative memory cells in a single modality cortex that functions as the core to encode and translate the associated signals among multiple modalities as well as morphologically interconnect with other modality cortices. Strategies to test this hypothesis are presented below. A mouse model of associative learning was conducted by pairing whisker and odorant stimulations, whisker and tail stimulations, as well as whisker and gustatory stimulations sequentially. The formation of associative memory based on the core of the whisker signal was expectedly featured by reciprocal responses induced between whisker and olfactory signals, whisker and tail signals, as well as whisker and gustatory signals, similar to the cases about associative memory in previous studies (*Feng et al., 2017*; *Gao et al., 2019*). The interconnections of associative memory neurons among cross-modal cortices were examined by microinjecting anterograde and retrograde adeno-associated viruses (AAVs) carried genes for encoding fluorescent proteins into the barrel cortex and by detecting the expression of these fluorescent proteins in their interconnected

cortical areas, or the other way around. The recruitment of associative memory neurons was ensured by detecting their convergent reception of synapse contacts between fluorescent-expressed axon boutons from presynaptic neurons and fluorescent-labeled dendritic spines on postsynaptic neurons in local areas (*Feng et al., 2017*; *Gao et al., 2019*; *Lei et al., 2017*). The spike-encoding ability in response to these signals was used to ensure the recruitment of associative memory neurons functionally (*Feng et al., 2017*; *Wu et al., 2020*). The role of neuroligin-3 in the recruitment of associative memory neuron was investigated by applying short-hairpin RNA (shRNA) that are specific to silence neuroligin-3 mRNA (AAV-DJ/8-U6-mNlgn3) in the barrel cortex.

## Results

In this section, we present data about the recruitment of associative memory neurons in the core region correlated to the formation of associative memory and the role of neuroligin-3 in this process. The formation of associative memory was evoked by the pair-stimulations of the whisker signal with the odorant signal, the gustatory signal, and the tail signal sequentially. The associative memory neurons in the barrel cortex that was presumably the core region interconnected with piriform, gustatory, and S1-Tr cortices were functionally identified by in vivo electrophysiologically recording barrel cortical neurons in response to whisker, odorant, tail, and gustatory signals. The morphological identification of associative memory neurons in the barrel cortex was conducted by detecting convergent synapse innervations onto the barrel cortical neurons inputted from piriform, gustatory, and S1-Tr cortices alongside the interconnections of the barrel cortex with the piriform, gustatory, and S1-Tr cortices. The roles of neuroligin-3 in the new synapse formation and associative memory neuron recruitment were investigated by neuroligin-3 knockdown with shRNA specific for *neuroligin-3*.

### The formation of associative memory featured in the whisker signal as a core pattern

The associative learning was conducted by the pair-stimulations of the whisker tactile signal with the olfactory butyl acetate signal, the whisker tactile signal with the sucrose taste signal, and the whisker tactile signal with the tail-heating signal sequentially in the paired-stimulus group (PSG) of C57BL/6JThy1-YFP, in comparison with the unpaired-stimulus group (UPSG) of the same strain mice, for 12 days (*Figure 1A* and Materials and methods in detail). The formation of associative memory was accepted when the reciprocal retrievals of these associated signals emerge in PSG mice, such as odorant-induced whisker motion plus whisking-induced olfactory response, gustation-induced whisker motion plus whisking-induced gustatory response, as well as tail-heating-induced whisker motion plus whisking-induced tail swing.

To examine the formation of associative memory after pairing the whisker signal and the olfactory signal, we analyzed the whisker fluctuation in response to the olfactory signal and the olfactory response to the whisker signal. The olfactory signal by giving butyl acetate appears to induce whisker fluctuations in PSG mice (an example in the middle trace of left panel of *Figure 1B*), but not in UPSG mice (an example in bottom trace). Whisker fluctuation amplitudes, or whisking angles (degree), are 36.22±2.46 in PSG mice (red bar in right panel of *Figure 1B*; n=13) and 10.20±2.7 in UPSG mice (blue bar; n=14, p<0.001, one-way ANOVA). This odorant-induced whisker motion indicates the retrieval of the whisker signal by olfactory signal, that is, the formation of associative memory in PSG mice. On the other hand, the whisker signal by stimulating mouse whiskers in a 'T' maze appears to induce mice away from the butyl acetate due to the smelling of this odorant in PSG mice (an example in a top 'T' maze of left-top panel in *Figure 1C*), but not in UPSG mice (an example in a bottom 'T' maze of left-bottom panel). Percentages away from the butyl acetate block are 64.82 ± 2.8% in PSG mice (red bar in the right panel of *Figure 1C*; n=17) and 48.20 ± 4.62% in UPSG mice (blue bar; n=10, p<0.01, one-way ANOVA). This whisking-induced olfactory response indicates that the whisker signal induces the retrieval of the olfactory signal, or the formation of associative memory in PSG mice. Thus, the associative learning by pairing the whisker signal and the olfactory signal leads to the odorant-induced whisker motion and whisking-induced olfactory responses, that is, the reciprocal retrieval of the associated signals as a complete format of associative memory.

In the test of outcomes of associative learning by pairing the whisker signal and tail-heating signal, we analyzed the whisker fluctuation in response to the tail-heating signal and the tail swing in response

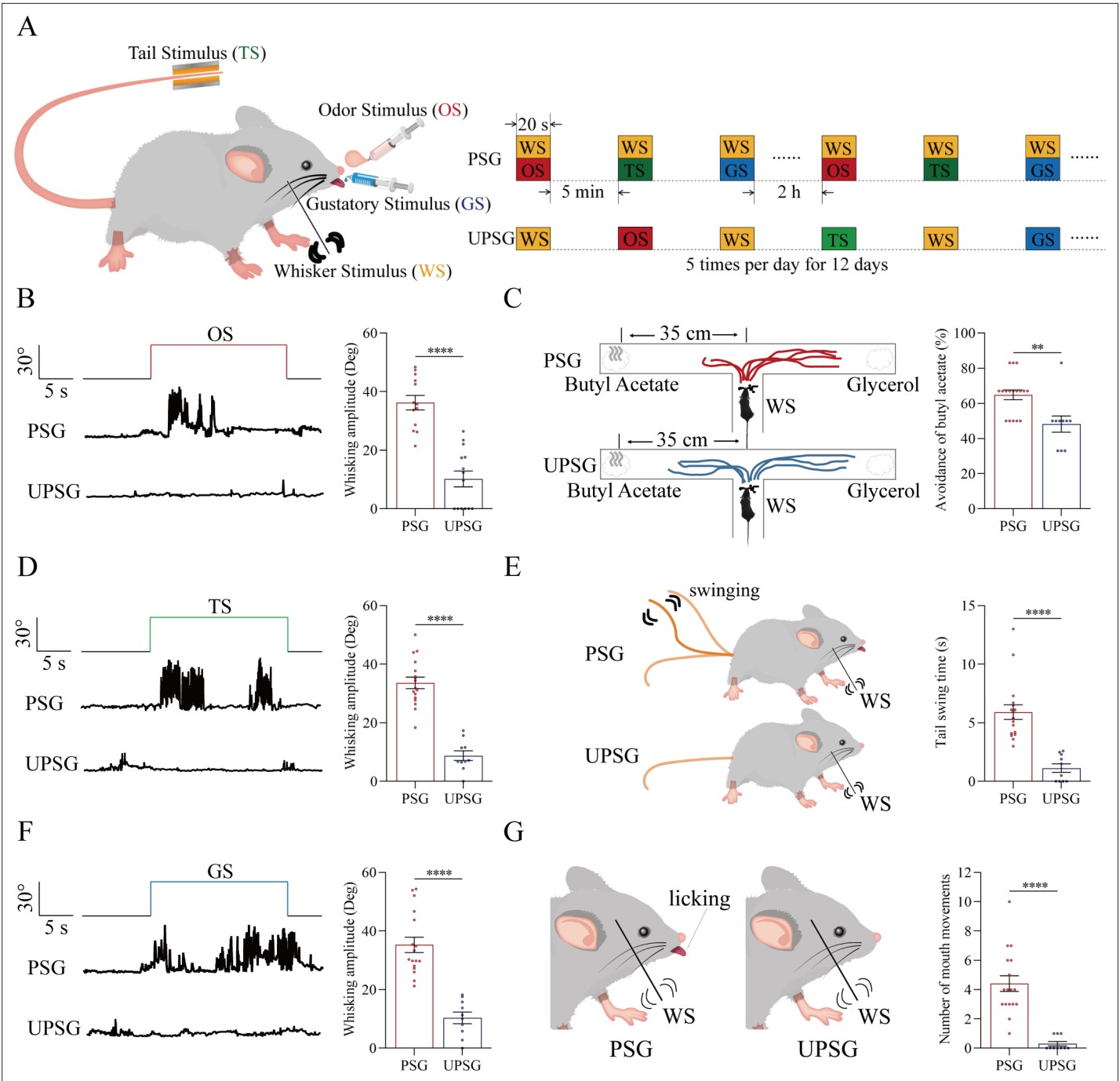

**Figure 1.** The pair-stimulations of the whisker stimulus (WS) with the odor stimulus (OS), the tail-heating stimulus (TS), and the gustatory stimulus (GS) sequentially lead to odorant-induced whisker motion versus whisking-induced olfactory response, tail-heating-induced whisker motion versus whisking-induced tail swing, and gustation-induced whisker motion versus whisking-induced taste response. (**A**) The associative learning in C57BL/6JThy1-YFP mice was conducted by the pair-stimulations of the whisker tactile signal with the olfactory butyl acetate signal, the whisker tactile signal with the sucrose taste signal, and the whisker tactile signal with the tail-heating signals sequentially in mice, which were assigned in paired-stimulus group (PSG), compared to mice in unpaired-stimulus group (UPSG), for 12 days. (**B, D, F**) The OS, the TS, and the GS appear to induce whisker motions in a fluctuation pattern in PSG mice, respectively, but not in UPSG mice. Calibration bars are 30°of whisker deflection and 5 s. The statistical analyses in right panels show whisking amplitudes in response to the OS, TS, and GS in PSG mice (red bar) and in UPSG mice (blue bar). (**C**) As indicated by the moving trace, PSG mice prefer to move away from the butyl acetate side as the olfactory response to the whisker stimulus compared with the UPSG mice (left panel). The percentage of avoidance of butyl acetate in PSG mice (red bar, right panel) and in UPSG mice (blue bar). (**E**) The WS appears to induce tail swing after training in PSG, but not in UPSG mice (left panel). The statistical analyses about tail swing time in response to the WS in PSG mice (red bar, right panel) and in UPSG mice (blue bar). (**G**) The WS appears to induce mouth movements after training in PSG mice, but not in UPSG mice (left panel). The statistical analyses about number of mouth movements in response to the WS in PSG mice (red bar, right panel) and in UPSG mice (blue bar).

to the mechanical whisker stimulation. The tail stimulation appears to induce whisker fluctuations in PSG mice (a sample in the middle trace of left panel in *Figure 1D*), but not in UPSG mice (a sample in bottom trace). Whisker fluctuation amplitudes are 33.63±1.95 in PSG mice (red bar in right panel of *Figure 1D*; n=17) and 8.79±1.66 in UPSG mice (blue bar; n=10, p<0.0001, one-way ANOVA). This tailing-induced whisker motion indicates the retrieval of the whisker signal by the tailing signal, that is, the formation of associative memory in PSG mice. On the other hand, the whisker signal by stimulating mouse whiskers appears to induce the tail swing in PSG mice (a sample in the left-top panel of *Figure 1E*), but not UPSG mice (a sample in left-top panel). The durations of the tail swing in response to the whisker stimulus are 5.91±0.63 s in PSG mice (red bar in right panel of *Figure 1E*; n=17) and 1.13±0.36 s in UPSG mice (blue bar; n=10, p<0.001, one-way ANOVA). This whisking-induced tail swing indicates the retrieval of the tail signal by the whisker signal, that is, the formation of associative memory in PSG mice. Thus, the associative learning by pairing the whisker signal and the tail signal leads to tailing-induced whisker motion and whisking-induced tail swing, or the reciprocal retrievals of these associated signals as a complete format of associative memory.

We also tested outcomes of associative learning by pairing the whisker signal and gustatory signal, in which the whisker fluctuation in response to the sucrose signal and the gustatory licking in response to the whisker stimulation were analyzed. The gustatory sucrose signal appears to induce the whisker fluctuation in PSG mice (a sample in the middle trace of left panel in *Figure 1F*), but not in UPSG mice (a sample in bottom trace). Whisker fluctuation amplitudes are 35.27±2.62 in PSG mice (red bar in right panel of *Figure 1F*; n=17) and 10.33±1.96 in UPSG mice (blue bar; n=10, p<0.0001, one-way ANOVA). This gustation-induced whisker motion indicates the retrieval of the whisker signal by gustatory signal, that is, the formation of associative memory in PSG mice. On the other hand, the whisker signal by stimulating mouse whiskers appears to evoke tongue-licking mouth lips in PSG mice (a sample in the left-top panel in *Figure 1G*), but not UPSG mice (a sample in left-top panel). The times of tongue licking-out in response to whisker stimulus are 4.41±0.54 in PSG mice (red bar in right panel of *Figure 1G*; n=17) and 0.3±0.15 in UPSG mice (blue bar; n=10, p<0.0001, one-way ANOVA). The whisking-induced gustatory response implies the retrieval of the gustatory signal by the whisker signal, or the formation of associative memory in PSG mice. Therefore, the associative learning by pairing the whisker signal and the gustatory signal leads to gustation-induced whisker motion and whisking-induced gustatory response, or the reciprocal retrievals of these associated signals as a complete format of associative memory.

These types of associative memory are featured by the reciprocal retrievals of the associated signals, in which the whisker signal as a core is associated with the olfactory signal, the tail signal, and the gustatory signal. As our general knowledge, the barrel cortex encodes the whisker signal, the piriform cortex encodes the olfactory signal, the S1-Tr cortex encodes the tail signal, and the gustatory cortex encodes the taste signal (*Brecht, 2007*; *Olavarria et al., 1984*; *Ogawa, 1994*; *Petersen, 2003*; *Wilson, 2001*). These reciprocal forms of associative memory are likely based on a process that the barrel cortex interconnects the piriform cortex, the S1-Tr cortex, and the gustatory cortex, after the learning of such associated signals has been done and the memories to these associated signals are formed, as implied in previous studies (*Feng et al., 2000*; *Wang et al., 2015*; *Gao et al., 2019*; *Liu et al., 2017*; *Yan et al., 2016*). If it is a case, we expect to observe that barrel cortical neurons mutually innervate cortical neurons in the piriform, S1-Tr, and gustatory cortices as well as convergently receive synapse innervations from these cortices by neural tracing. Moreover, these barrel cortical neurons become to encode all of these associated signals, that is, associative memory neurons of encoding multiple signals.

## Associative memory neurons are recruited in the barrel cortex for this core pattern

To study the interconnections of the barrel cortex with piriform, S1-Tr, and gustatory cortices, we microinjected AAV2/retro-CMV-EGFP and AAV2/8-CMV-tdTomato into the barrel cortex (*Figure 2A*). AAV2/8-CMV-tdTomato was uptaken and expressed in the somata of barrel cortical neurons, and tdTomato was transported to their axonal boutons and terminals in target regions in the anterograde manner. AAV2/retro-CMV-EGFP was uptaken by the axonal terminals and boutons of barrel cortical neurons, transported in the retrograde manner, and expressed in the somata of the neurons that projected to the barrel cortex. If the interconnections were formed between the barrel cortex

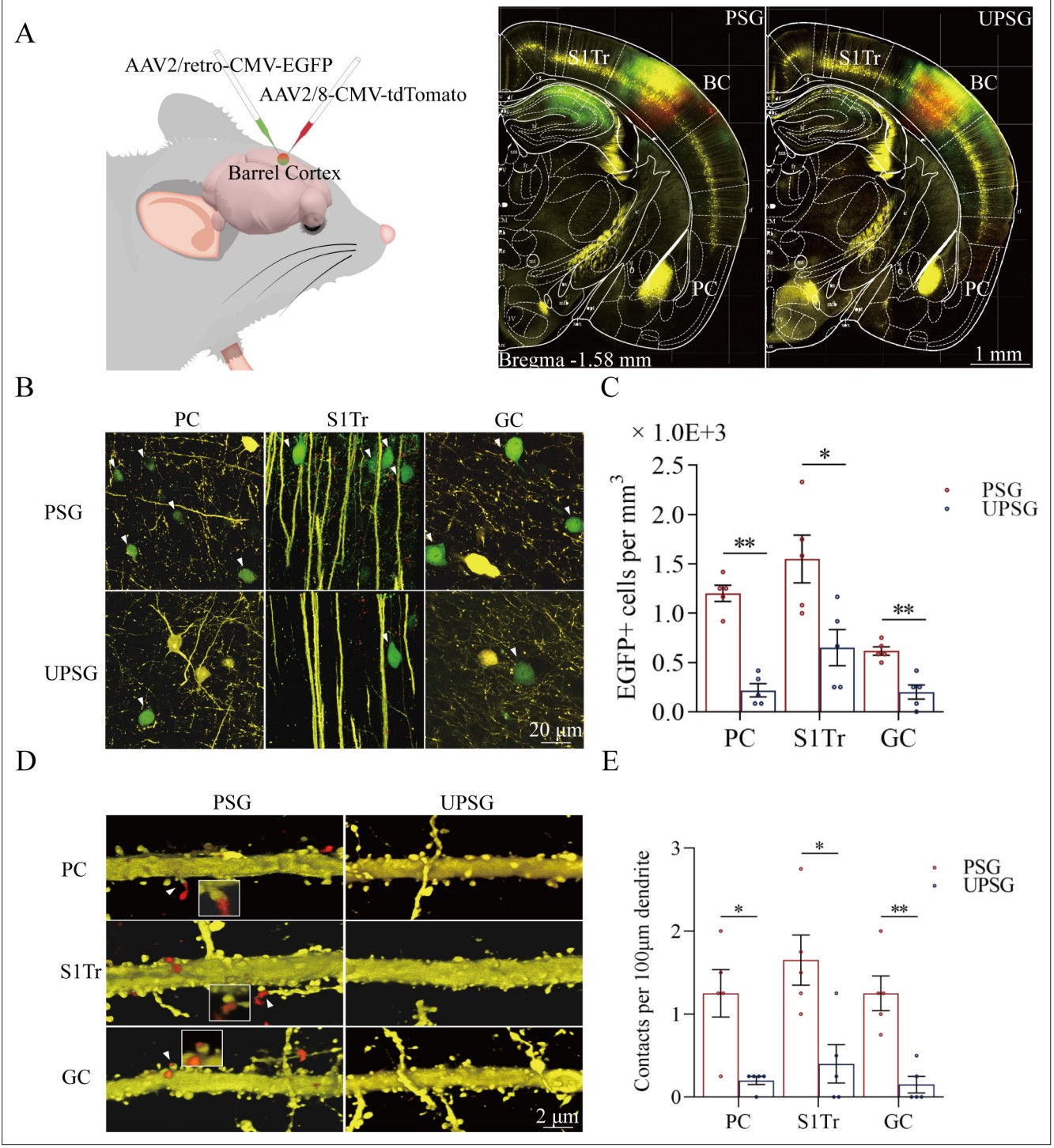

**Figure 2.** Associative learning by pairing whisker stimulus (WS) with the odor stimulus (OS), WS with tail-heating stimulus (TS), and WS with gustatory stimulus (GS) induces synapse innervations from PC, S1-Tr, and GC to BC, as well as from BC to PC, S1-Tr, and GC. (**A**) The AAV2/retro-CMV-EGFP and AAV2/8-CMV-tdTomato were microinjected into the barrel cortex. (**B**) EGFP-labeled neurons in the PC, S1-Tr, and GC in paired-stimulus group (PSG) mice and unpaired-stimulus group (UPSG) mice. (**C**) The densities of EGFP-labeled neurons in the PC, S1-Tr, and GC in PSG mice (red bar) and UPSG mice (blue bar). (**D**) Td-Tomato-labeled axonal boutons from BC and their contacts on the dendritic spine of PC, S1-Tr, and GC neurons in PSG mice and UPSG mice. (**E**) The densities of synapse contacts consisting of tdTomato-labeled boutons and yellow fluorescent protein (YFP)-labeled spines in the PC, S1-Tr, and GC in PSG mice (red bar) and UPSG mice (blue bar).

and piriform, S1-Tr, and gustatory cortices after associative learning and memory, we should detect tdTomato-labeled axonal boutons and EGFP-labeled somata in piriform, S1-Tr, and gustatory cortices. It is noteworthy that much retrograde transportation of EGFP is detected in hippocampal CA1-CA3 areas from the mice experienced associative learning (PSG mice), in comparison with UPSG mice (right panel in *Figure 2A*). This substantially increased synapse innervation from the hippocampus to the barrel cortex after memory formation grants a possibility that the hippocampus strengthens memories. Together with an observation about the secondary associative memory cells in the hippocampus innervated from the primary associative memory cells in the barrel cortex (our unpublished data), we suggest that memories to specific signals in the barrel cortex and other cortices are facilitated and strengthened by the interactions between the hippocampus and cerebral cortices.

*Figure 2B* illustrates GFP-labeled neurons within the piriform cortex (left panels), the S1-Tr cortex (middle panels), and the gustatory cortex (right panels) in PSG mice (top panels) and UPSG mice (bottom panels). The densities of GFP-labeled neurons from the piriform cortex, the S1-Tr cortex, and the gustatory cortex appear higher in PSG mice than in UPSG mice. The densities of GFP-labeled neurons (neurons per $mm^3$) in the piriform cortex are $1.20 \pm 0.08 \times 10^3$ /$mm^3$ in PSG (red bar in *Figure 2C*; n=15 cubes from 5 mice) and $0.22 \pm 0.07 \times 10^3$/$mm^3$ in UPSG (blue bar, n=15 cubes from 5 mice; p<0.01, one-way ANOVA). GFP-labeled neurons per $mm^3$ in the S1-Tr cortex are $1.55 \pm 0.24 \times 10^3$ in PSG (red bar in *Figure 2C*; n=15 cubes from 5 mice) and $0.65 \pm 0.18 \times 10^3$ in UPSG (blue bar, n=15 cubes from 5 mice; p<0.05, one-way ANOVA). GFP-labeled neurons per $mm^3$ in the gustatory cortex are $0.62 \pm 0.04 \times 10^3$ in PSG (red bar in *Figure 2C*; n=15 cubes from 5 mice) and $0.20 \pm 0.07 \times 10^3$ in UPSG (blue bar; n=15 cubes from 5 mice, p<0.01, one-way ANOVA). These data indicate that neuronal axons from piriform, S1-Tr, and gustatory cortices project into the barrel cortex in a convergence manner after associative learning and memory.

*Figure 2D* illustrates the innervations of tdTomato-labeled axon boutons (red dots) onto the spines of neuronal dendrites (yellow protrusion), or synapse contacts, in the piriform cortex (top panels), the S1-Tr cortex (middle panels), and the gustatory cortex (bottom panels) from the PSG mice (left panels) and UPSG mice (right panels). Synapse contacts on neuronal dendrites within piriform, S1-Tr, and the gustatory cortices appear higher in PSG mice than in UPSG mice. Synapse contacts per 100 μm dendrite in the piriform cortex are $1.25 \pm 0.18 \times 10^3$ in PSG (red bar in *Figure 2E*, n=20 dendrites from 5 mice) and $0.15 \pm 0.08 \times 10^3$ in UPSG (blue bar, n=20 dendrite from 5 mice; p<0.05, one-way ANOVA). Synapse contacts per 100 μm dendrite in the S1-Tr cortex are $1.65 \pm 0.3 \times 10^3$ in PSG (red bar in *Figure 2E*, n=20 dendrites from 5 mice) and $0.40 \pm 0.23 \times 10^3$ in UPSG (blue bar, n=20 dendrite from 5 mice; p<0.05, one-way ANOVA). Synapse contacts per 100 μm dendrite in the gustatory cortex are $1.25 \pm 0.21 \times 10^3$ in PSG (red bar in *Figure 2E*, n=20 dendrite from 5 mice) and $0.15 \pm 0.10 \times 10^3$ in UPSG (blue bar, n=20 dendrite from 5 mice; p<0.01, one-way ANOVA). Such results indicate that barrel cortical neurons project their axons into piriform, S1-Tr, and gustatory cortices and make new synapses onto piriform, S1-Tr, and gustatory cortical neurons after the associative learning and memory.

We further examined whether these synapse interconnections formed between the barrel cortex and piriform, S1-Tr and gustatory cortices after the associative learning and memory had recruited associative memory neurons by morphological and functional approaches. If it is a case, we expect to detect that barrel cortical neurons receive convergent synapse innervations from piriform, S1-Tr, and gustatory cortices as well as are able to encode olfactory, tail, and gustatory signals from these inputs along with the whisker signal from the thalamus after the associative learning and memory.

Morphological evidence about convergent synapse innervations on barrel cortical neurons from PSG mice was presented in *Figure 3*. *Figure 3A* illustrates the microinjections of AAV-CMV-fluorescents into the piriform cortex (tdTomato), the S1-Tr cortex (GFP), and the gustatory cortex (BFP) in PSG mice (left panels) and in UPSG mice (right panels), respectively. *Figure 3B* illustrates tdTomato-, GFP-, and BFP-labeled axonal boutons in the barrel cortex from PSG mice (top panel), in comparison with those from UPSG mice (bottom panel). The densities of tdTomato-, GFP-, and BFP-labeled boutons in the barrel cortex appear higher in PSG mice than in UPSG mice. *Figure 3C* shows the statistical analyses about the densities of axon boutons (boutons per $mm^3$) in the two groups of mice. The densities of tdTomato-labeled boutons in the barrel cortex are $1.61 \pm 0.14 \times 10^4$/$mm^3$ in PSG (red bar in *Figure 3C*, n=15 cubes from 5 mice) and $0.43 \pm 0.05 \times 10^4$/$mm^3$ in UPSG (blue bar, n=15 cubes from 5 mice; p<0.01, one-way ANOVA). The densities of GFP-labeled axon boutons in

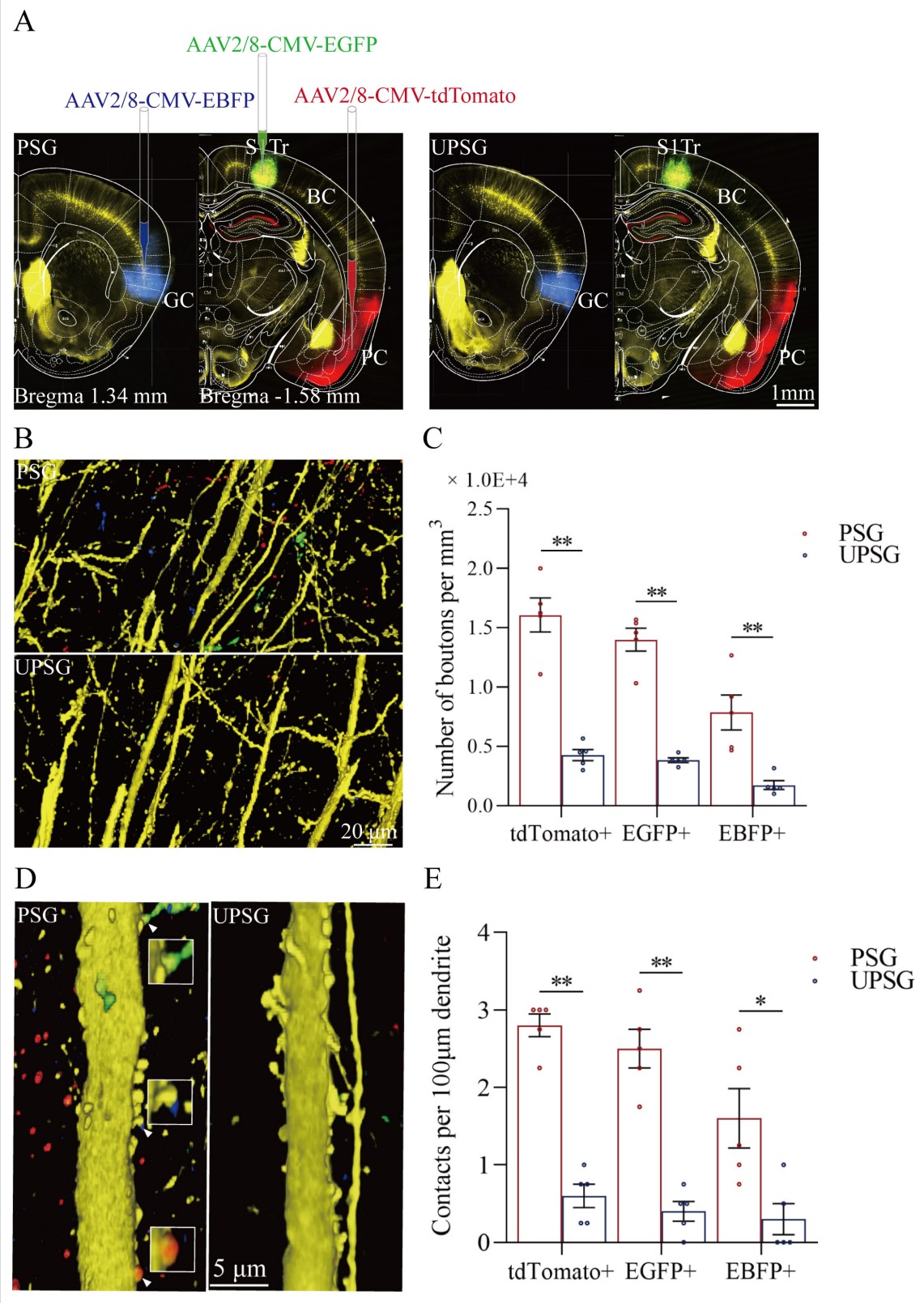

**Figure 3.** The convergent synapse innervations of neurons from the piriform cortex, gustatory cortex, and S1-Tr cortex onto barrel cortical neurons. (**A**) Neuronal tracing was done by injecting AAV2/8-CMV-EBFP into the GC, AAV2/8-CMV-tdTomato into the PC, and AAV2/8-CMV-EGFP into the S1-Tr, and by detecting their presence in the BC. (**B**) The axon boutons labeled by EBFP, EGFP, and tdTomato are detected in the BC of paired-stimulus group (PSG) mice (top panel), compared with those in unpaired-stimulus group (UPSG) mice (bottom panel). (**C**) The densities of EBFP-labeled, EGFP-

*Figure 3 continued on next page*

*Figure 3 continued*

labeled, and tdTomato-labeled boutons in the BC in PSG mice (red bar) and in UPSG mice (blue bar). (**D**) Synapse contacts between the spines of yellow fluorescent protein (YFP)-labeled glutamatergic neurons and the axon boutons labeled by EBFP, EGFP, or tdTomato are detected in the BC of PSG mice (left panel), compared with those in UPSG (right panel). (**E**) The densities of synapse contacts between the spines of YFP-labeled neurons and the axon boutons labeled by EBFP, EGFP, or tdTomato in PSG mice (red bar) and in UPSG mice (blue bar).

the barrel cortex are $1.40\pm0.10 \times 10^4/mm^3$ in PSG (red bar in *Figure 3C*, n=15 cubes from 5 mice) and $0.38\pm0.02 \times 10^4 /mm^3$ in UPSG (blue bar, n=15 cubes from 5 mice; p<0.01, one-way ANOVA). The densities of BFP-labeled axon boutons in the barrel cortex are $0.79\pm0.15 \times 10^4/mm^3$ in PSG (red bar in *Figure 3C*, n=15 cubes from 5 mice) and $0.17\pm0.04 \times 10^4/mm^3$ in UPSG (blue bar, n=15 cubes from 5 mice; p<0.01, one-way ANOVA). Therefore, barrel cortices in mice that express associative memory receive more axon projections convergently from piriform, S1-Tr, and gustatory cortices.

*Figure 3D–E* shows the analysis about the density of synapse contacts on the dendritic spines of barrel cortical neurons. New synapses formed on barrel cortical neurons in PSG mice are the contacts between tdTomato-, GFP-, and BFP-labeled axon boutons from presynaptic neurons and yellow fluorescent protein (YFP)-labeled postsynaptic dendritic spines of barrel cortical neurons. Dendritic synapse contacts on barrel cortical neurons appear to emerge in PSG mice (left panel in *Figure 3D*), but not in UPSG mice (right panel). Statistical analyses about the densities of synapse contacts on barrel cortical neurons (synapse contacts per 100 μm dendrite) are presented in *Figure 3E*. The densities of tdTomato-labeled synapse contacts on barrel cortical neurons are $2.80\pm0.15$ in PSG (red bar in *Figure 3E*, n=20 dendrites from 5 mice) and $0.60\pm0.15$ in UPSG (blue bar, n=20 dendrites from

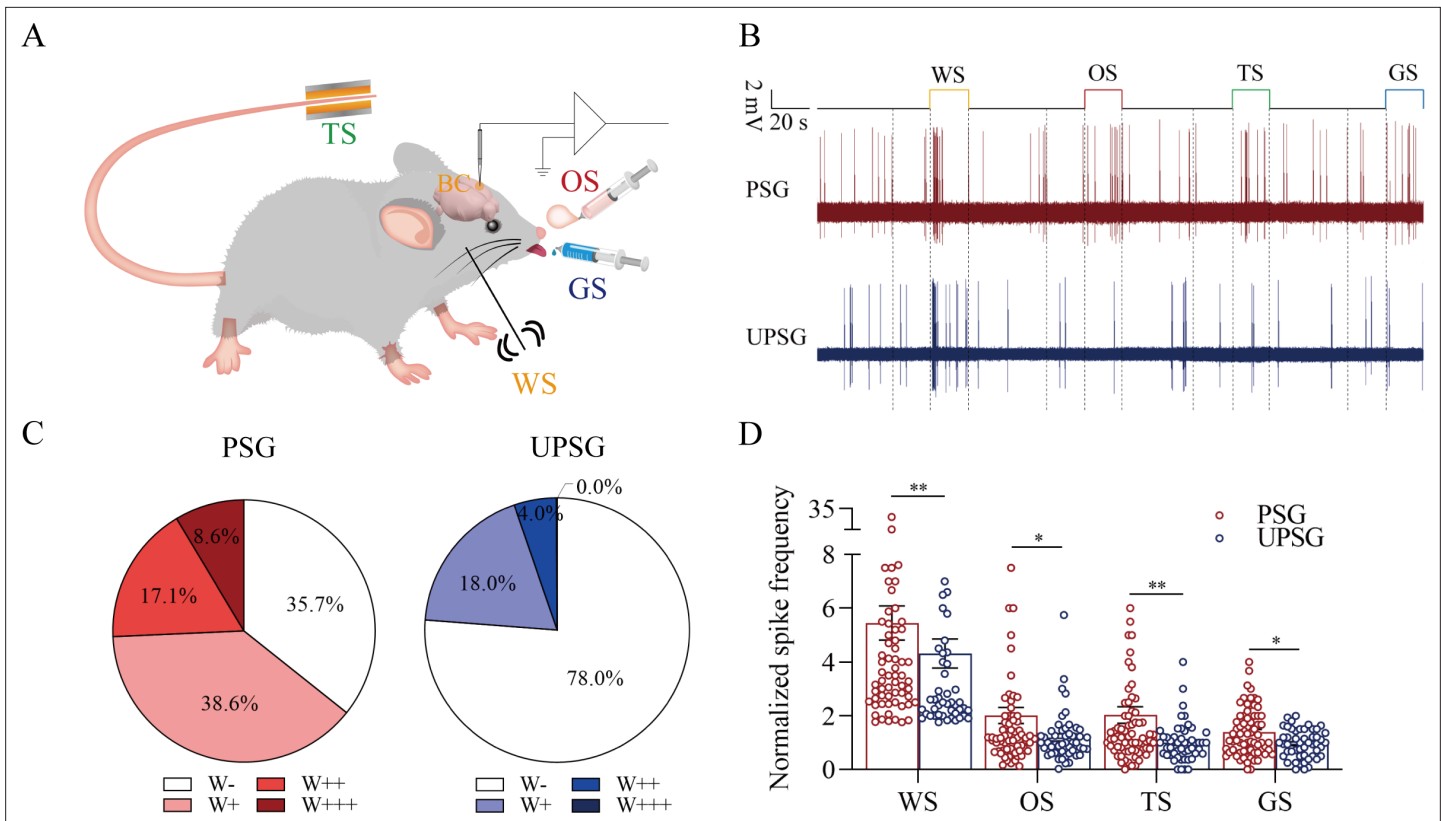

**Figure 4.** The barrel cortical neurons were responding to odor, tail-heating, and gustatory sucrose signals alongside the whisker tactile signal in paired-stimulus group (PSG) mice. (**A**) A diagram illustrates a recording of local field potentials (LFPs) in the barrel cortex. (**B**) An example of barrel cortical neuron in response to butyl acetate, tail-heating, sucrose, and whisker tactile signals from PSG mouse (red trace) and unpaired-stimulus group (UPSG) mouse (blue trace). The calibration bars are 2 mV and 20 s. (**C**) The percentages of associative neurons in PSG mice (left panel) and in UPSG mice (right panel). Neurons that respond only to whisker stimulus (WS) are labeled as W-, and those that respond to WS and one, two, or all three other signals are labeled as W+, W++, and W+++, respectively. (**D**) The normalized spike frequencies in response to whisker tactile signals, butyl acetate, tail-heating signal, and sucrose signal in PSG (red bar) and in UPSG (blue bar), respectively.

5 mice; p<0.01, one-way ANOVA). The densities of GFP-labeled synapse contacts on barrel cortical neurons are 2.50±0.25 in PSG (red bar in *Figure 3E*, n=20 dendrites from 5 mice) and 0.40±0.13 in UPSG (blue bar, n=20 dendrites from 5 mice; p<0.01, one-way ANOVA). The densities of BFP-labeled synapse contacts on barrel cortical neurons are 1.60±0.38 in PSG (red bar in *Figure 3E*, n=20 dendrites from 5 mice) and 0.30±0.20 in UPSG (blue bar, n=20 dendrites from 5 mice; p<0.05, one-way ANOVA). Therefore, barrel cortical neurons in the mice that express associative memory receive more new synapse contacts from piriform, S1-Tr, and gustatory cortices. Data in *Figure 3* indicate morphological evidence about the recruitment of associative memory neurons in the barrel cortex.

The functional evidence about barrel cortical neurons encoding odor, gustatory sucrose, and tail-heating signals alongside the whisker signal in PSG mice after their associative learning, that is, the recruitment of associative memory neurons, is presented in *Figure 4*. The responses of barrel cortical neurons to the butyl acetate, tail-heating, or sucrose signals alongside the whisker signal were examined by recording their activities induced by these signals (*Figure 4A*). If barrel cortical neurons encoded more signals besides the whisker tactile signal, they were presumably called as associative memory neurons (*Feng et al., 2017*; *Wang et al., 2015*; *Gao et al., 2016*; *Wang et al., 2019b*; *Wu et al., 2020*). *Figure 4B* illustrates the recording samples of barrel cortical neurons in response to butyl acetate, tail-heating, sucrose and whisker tactile signals from a PSG mouse (red trace) and a UPSG mouse (blue), respectively. Barrel cortical neurons in PSG mouse can respond to the butyl acetate, tail-heating, and sucrose signals alongside the whisker tactile signal, in comparison with those in UPSG mouse. The percentages of barrel cortical neurons in response to the whisker signal plus butyl acetate, tail-heating and/or sucrose signals are 64.3% in PSG mice and 22% in UPSG mice (*Figure 4C*; p<0.01, $\chi^2$-test). Thus, associative memory neurons are functionally recruited in the barrel cortex after the associative learning. Furthermore, based on the analysis of neuronal spike frequencies in response to these signals, or the activity strength of associative memory neurons, normalized spike frequencies in response to butyl acetate are 2.03±0.3 in PSG (red bar in *Figure 4D*; n=70 neurons from 12 mice) and 1.19±0.13 in UPSG (blue bar, n=50 neurons from 8 mice, p<0.05, one-way ANOVA). The activity strengths in response to the tail-heating signal are 2.03±0.31 in PSG (red bar in *Figure 4D*, n=70 neurons from 12 mice) and 1.01±0.1 in UPSG (blue bar, n=50 neurons from 8 mice; p<0.01, one-way ANOVA). The activity strengths in response to the sucrose signal are 1.39±0.11 in PSG (red bar in *Figure 4D*, n=70 neurons from 12 mice) and 0.96±0.07 in UPSG (blue bar, n=50 cells from 8 mice; p<0.05, one-way ANOVA), respectively. Therefore, the activity levels of associative memory neurons are strengthened. The data above indicate that the associative learning and memory by pairing the whisker signal with olfactory, tail, and gustatory signals recruit associative memory neurons in the barrel cortex that encode these signals.

## Associative memory neurons in the barrel cortex as a core station to facilitate signal retrievals

Associative memory cells in the barrel cortex as the core of morphological interconnection with piriform, S1-Tr, and gustatory cortices and functional interaction with these cortical regions leads to the joint storage and the reciprocal retrieval of the whisker signal with olfactory, tail, and gustatory signals associated during the associative learning. The interconnections of the barrel cortex with piriform, S1-Tr, and gustatory cortices may also enable the barrel cortex to be a core linkage among piriform, S1-Tr, and gustatory cortices for their indirect interconnections as well as to transfer the signals among piriform, S1-Tr, and gustatory cortices for their indirect interactions (*Figure 5A*). In this regard, the activity of the piriform cortex may activate the barrel cortex and in turn indirectly activate Sr-Tr and gustatory cortices, or the other way around. The olfactory signal may retrieve the whisker signal and then indirectly retrieve somatic and gustatory signals, or the other way around. This possibility has been examined, as presented in *Figure 5B–D*.

The tail-heating or sucrose appears to induce the olfactory responses (left panel in *Figure 5B*) besides whisker fluctuations (*Figure 1*) in PSG mice, in which the tail-heating or the sucrose can induce the mice away from butyl acetate due to their smelling this odorant (examples in top and bottom 'T' mazes, respectively), but not in UPSG mice. Percentages away from the butyl acetate block by the tail-heating signal are 69.59 ± 4.08% in PSG mice (red bar in the left panel of *Figure 5C*; n=17) and 48.30 ± 4.7% in UPSG mice (blue bar; n=10; p<0.01, one-way ANOVA). Percentages away from the butyl acetate block induced by the sucrose are 66.59 ± 4.73% in PSG mice (red bar in the right panel

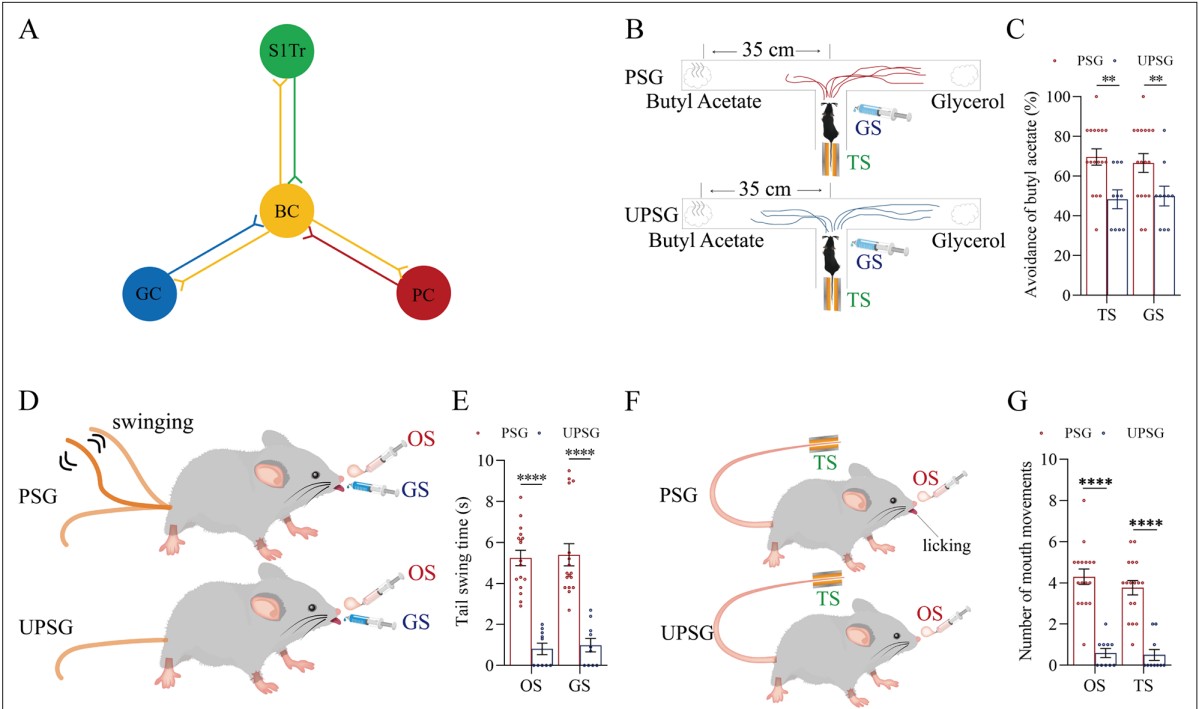

**Figure 5.** The interconnections of the barrel cortex with the piriform, S1-Tr, and gustatory cortices enable this core of the barrel cortex constitute a linkage among piriform, S1-Tr, and gustatory cortices for the piriform cortex, the S1-Tr cortex, and the gustatory cortex to be indirectly interconnected. (**A**) The barrel cortex can become the core station for the first-order and secondary-order associative memory. (**B**) The tail-heating or sucrose appears to induce the olfactory responses in paired-stimulus group (PSG) mice, in which the tail-heating or the sucrose induces mice away from the butyl acetate due to their smelling this odorant (examples in top and bottom 'T' mazes, respectively), but not in unpaired-stimulus group (UPSG) mice. (**C**) Percentages away from the butyl acetate block by the tail-heating and the sucrose in PSG mice (red bar) and in UPSG mice (blue bar). (**D**) The butyl acetate or the sucrose appears to induce the tail swing besides the whisker fluctuation in PSG mice, but not in UPSG mice. (**E**) The durations of tail swing by the butyl acetate and the sucrose in PSG mice (red bar) and in UPSG mice (blue bar). (**F**) The butyl acetate or the tail-heating appears to induce the gustatory responses besides the whisker fluctuation in PSG mice, but not in UPSG mice. (**G**) The times of tongue-out licking by the butyl acetate and the tail-heating in PSG mice (red bar) and in UPSG mice (blue bar).

of *Figure 5C*; n=17) and 49.90 ± 4.99% in UPSG mice (blue bar; n=10; p<0.01, one-way ANOVA). In addition, the butyl acetate or the sucrose appears to induce the tail swing (top panel in *Figure 5D*) besides whisker fluctuations (*Figure 1*) in PSG mice, but not UPSG mice. The durations of tail swing induced by the butyl acetate are 5.24±0.37 s in PSG mice (red bar in the left panel of *Figure 5E*; n=17) and 0.81±0.28 s in UPSG mice (blue bar; n=10; p<0.0001, one-way ANOVA). The durations of tail swing induced by the sucrose are 5.40±0.54 s in PSG mice (red bar in right panel of *Figure 5E*; n=17) and 0.99±0.33 s in UPSG mice (blue bar; n=10; p<0.0001, one-way ANOVA). Moreover, the butyl acetate or the tail-heating appears to induce gustatory responses (top panel in *Figure 5F*) besides whisker fluctuations (*Figure 1*) in PSG mice, but not UPSG mice. The times of tongue-out licking induced by the butyl acetate are 4.29±0.37 in PSG mice (red bar in the left panel of *Figure 5G*; n=17) and 0.6±0.22 in UPSG mice (blue bar; n=10; p<0.0001, one-way ANOVA). The times of tongue-licking mouth lips induced by the tail-heating are 3.77±0.35 s in PSG mice (red bar in right panel of *Figure 5G*; n=17) and 0.50±0.27 in UPSG mice (blue bar; n=10; p<0.0001, one-way ANOVA). These data imply that the activation to one of piriform, S1-Tr, and gustatory cortices by their innate-encoded signals is able to activate the rest of them indirectly by jumping over the barrel cortex. Neural bases for the retrieval of these signals indirectly may be due to the interconnections of the barrel cortex with piriform, S1-Tr, and gustatory cortices (*Figure 5A*), in which the barrel cortex as a hub eases the signal retrievals among the interconnected cerebral cortices. This indication has been further examined by blocking the function of the barrel cortex below.

In terms of molecular mechanisms underlying the recruitment of associative memory cells, certain molecules in relevance to axonal prolongation and synapse formation are essential (*Feng et al., 2017*; *Lei et al., 2017*; *Wu et al., 2020*). If the synapse formation is required for the recruitment of

associative memory neurons, the linkage of presynaptic boutons and postsynaptic spines is expectedly needed. Neuroligin-3 as one of the linkage proteins (*Craig and Kang, 2007*; *Li et al., 2022*; *Lisé and El-Husseini, 2006*; *Südhof, 2017*; *Uchigashima et al., 2021*) for synapse fixations has been examined, in which short-chain RNA (shRNA) specific for *neurolingin-3* (*Chang et al., 2006*; *Khatri et al., 2012*; *Pardridge, 2007*; *Pushparaj et al., 2008*; *Rao et al., 2009*) was microinjected into the barrel cortex. Subsequently after the associative learning paradigms, the formation of associative memory, the innervation of new synapses, as well as the recruitment of associative memory neurons in the barrel cortex were examined by behavioral tasks, neural tracing morphologically and electrophysiological recording functionally.

## Neuroligin-3 is required for associative memory formation and memory cell recruitment

If new synapses are formed by the linkage of presynaptic boutons and postsynaptic spines, the downregulation of *neuroligin-3* by shRNA is expected to prevent the formation of associative memory, the innervation of new synapses, and the recruitment of associative memory neurons (*Figure 6A*). shRNA specific for neuroligin-3 was carried by AAV-D/J8-U6-mNlgn3-EGFP (*Figure 6B*). *Neurolingin-3* shRNA and its scramble control were microinjected into barrel cortices in two subgroups of mice within the categories of PSG group, that is, shRNA subgroup and shRNA scramble subgroup, before the associative learning. After the associative learning by pairing whisker and olfactory signals, whisker and tail-heating signals, as well as whisker and gustatory signals serially for 2 weeks in PSG mice (Materials and methods), we investigated the expression of associative memory in behavior tasks, the formation of new synapses in morphology, and the recruitment of associative memory neurons by functional and morphological approaches. The paradigm of the associative learning and memory was similar to those in *Figure 1A–B*.

Behavioral tasks about the effect of neuroligin-3 knockdown on the formation of associative memory directly and indirectly are presented in *Figure 6C–F*. After the paradigm of associative learning was processed to neuroligin-3 knockdown mice and shRNA scramble control mice, the amplitudes of odorant-induced whisker motion are 20.11±0.8 in neuroligin-3 knockdown mice (blue bar in *Figure 6C*, n=10) and 32.11±0.65 in shRNA scramble control mice (red bar, n=10; p<0.0001, one-way ANOVA). The amplitudes of tailing-induced whisker motion are 18.62±1.26 in neuroligin-3 knockdown mice (blue bar in *Figure 6C*, n=10) and 32.54±1.12 in shRNA scramble control mice (red bar, n=10; p<0.0001, one-way ANOVA). The amplitudes of sucrose-induced whisker motion are 20.22±0.9 in neuroligin-3 knockdown mice (blue bar in *Figure 6C*, n=10) and 32.54±1.12 in shRNA scramble control mice (red bar, n=15; p<0.0001, one-way ANOVA). On the other hand, the rates of avoidance to butyl acetate in whisking-induced olfactory responses are 55.63 ± 3.55% in neuroligin-3 knockdown mice (blue bar in the left panel of *Figure 6D*; n=15) and 68.93 ± 5.07% in shRNA scramble control mice (red bar; n=15, p<0.05, one-way ANOVA). The durations of whisking-induced tail swing are 1.56±0.34 s in neuroligin-3 knockdown mice (blue bar in *Figure 6D*, middle panel; n=10) and 4.60±0.37 s in shRNA scramble mice (red bar; n=10, p<0.0001, one-way ANOVA). The times of whisking-induced lip lickings are 1.00±0.26 in neuroligin-3 knockdown mice (blue bar in right panel of *Figure 6D*; n=10) and 4.20±0.36 in shRNA scramble control mice (red bar; n=10, p<0.0001, one-way ANOVA). These results indicate that neuroligin-3 knockdown downregulates the joint storage and the reciprocal retrieval of the associated signals, in which the whisker signal is a core signal and the barrel cortex is a core place for these processes since neuroligin-3 knockdown has been executed in the barrel cortex.

Furthermore, the data about the reciprocal retrievals among the olfactory signal, the tailing signal, and the gustatory signal indirectly by the linkage of the barrel cortex where neuroligin-3 has been knockdown are presented in *Figure 6E–G*. *Figure 6E* shows the influences of tail-heating and sucrose signals on olfactory responses in those two subgroups of barrel cortical neuroligin-3 knockdown and shRNA scramble control mice. The rates of the avoidance to butyl acetate induced by tail-heating and sucrose signals are 52.20 ± 3.62% (blue bar in left sides of *Figure 6E*) and 53.33 ± 3.75% (blue bar in right sides) in neuroligin-3 knockdown mice (n=15), in comparison with the values 66.87 ± 5.1% (red bar in left sides of *Figure 6E*) and 66.67 ± 3.97% (red bar in right sides) in shRNA scramble control mice (n=15, p<0.05, one-way ANOVA). *Figure 6F* illustrates the influences of butyl acetate and sucrose signals on tail swing in two subgroups of barrel cortical neuroligin-3 knockdown and

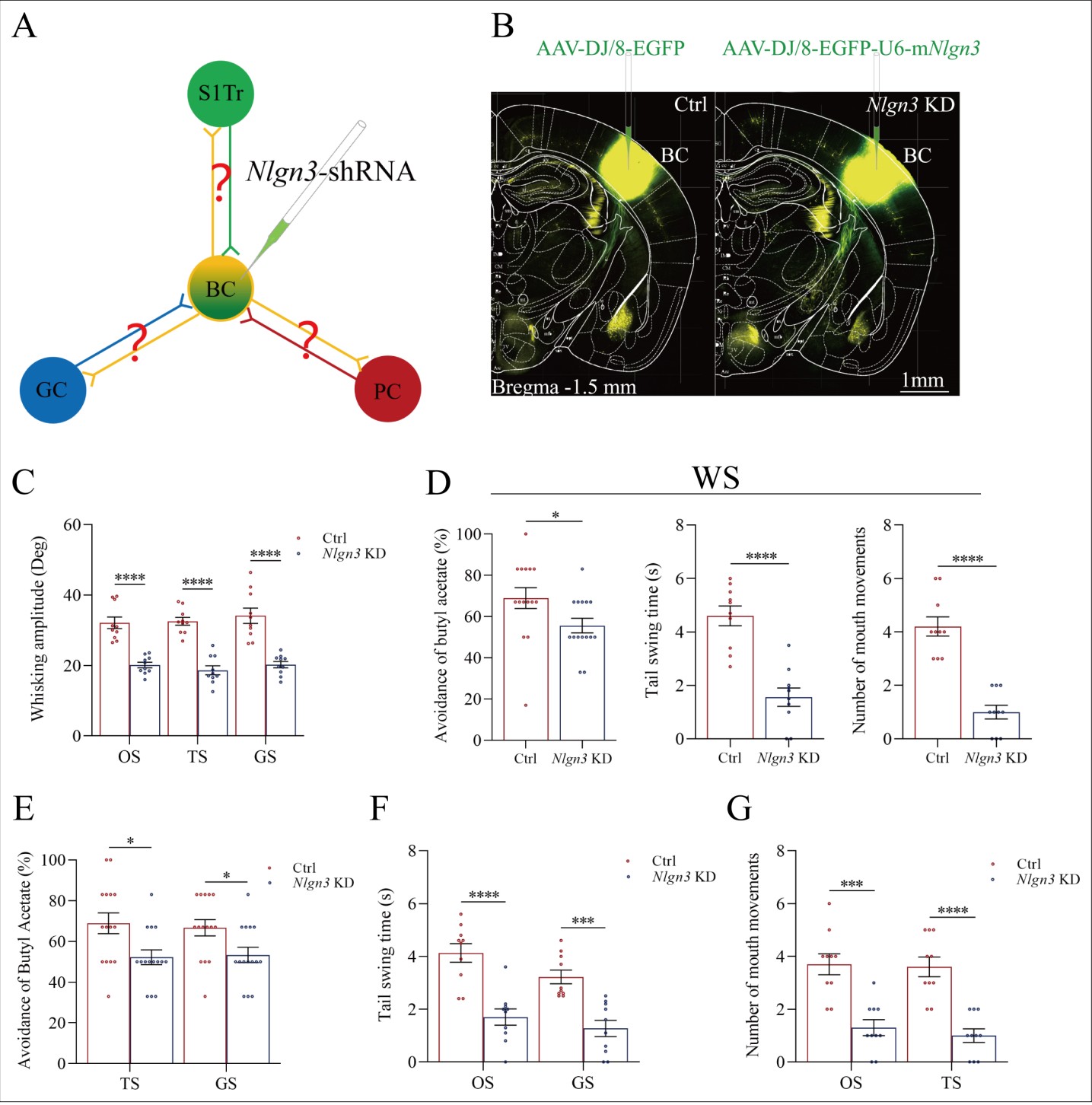

**Figure 6.** Neuroligin-3 knockdown significantly downregulates the joint storage and the reciprocal retrieval of the associated signals. (**A**) The downregulation of neuroligin-3 through short-hairpin RNA (shRNA) is expected to suppress the innervation of new synapses as well as the recruitment of associative memory neurons in the BC. (**B**) shRNA specific for neurolingin-3 and its scramble control RNA were microinjected into the barrel cortices in shRNA group and shRNA scramble subgroups. (**C**) The amplitudes of whisker motion induced by the odor stimulus (OS), tail-heating stimulus (TS), and gustatory stimulus (GS) in scramble group (red bar) and sh*Nlgn3* group (blue bar). (**D**) The left panel shows the rates of avoidance to the butyl acetate in whisking-induced olfactory response in neuroligin-3 knockdown mice (blue bar) and in shRNA scramble control mice (red bar). The middle panel shows the durations of whisking-induced tail swing in neuroligin-3 knockdown mice and in shRNA scramble control mice. The right panel shows the times of whisking-induced lip lickings in neuroligin-3 knockdown mice and in shRNA scramble control mice. (**E**) Percentages away from the butyl acetate block by the tail-heating and the sucrose in shRNA scramble control mice (red bar) and in neuroligin-3 knockdown mice (blue bar). (**F**) The durations of tail swing

*Figure 6 continued on next page*

*Figure 6 continued*

by the butyl acetate and the sucrose in shRNA scramble control mice (red bar) and in neuroligin-3 knockdown mice (blue bar). (**G**) The times of tongue-out licking by the butyl acetate and the tail-heating in shRNA scramble control mice (red bar) and in neuroligin-3 knockdown mice (blue bar).

shRNA scramble control mice. The durations of the tail swing in response to the olfactory signal and the gustatory signal are 1.70±0.31 s (blue bar in left sides of *Figure 6F*) and 1.27±0.31 s (blue bar in right sides) in neuroligin-3 knockdown mice (n=10), in comparison with the values 4.13±0.35 s (red bar in left sides of *Figure 6E*) and 3.22±0.26 s (red bar in right sides) in shRNA scramble control (n=10, p<0.001, one-way ANOVA). *Figure 6G* shows the influences of butyl acetate and tail-heating on gustatory responses in two subgroups of neuroligin-3 knockdown in the barrel cortex and shRNA scramble control mice. The times of lip lickings in response to the olfactory signal and the tail-heating signal are 1.30±0.3 (blue bar in left sides of *Figure 6G*) and 1.0±0.26 s (blue bar in right sides) in neuroligin-3 knockdown mice (n=10), in comparison with those values 3.70±0.4 (red bar in left sides of *Figure 6G*) and 3.60±0.37 (red bar in right sides) in shRNA scramble control (n=10, p<0.001, one-way ANOVA). Therefore, the downregulation of the reciprocal retrievals among the olfactory signal, the tailing signal, and the gustatory signal by neuroligin-3 knockdown in the barrel cortex indicates that the indirect interaction among piriform, S1-Tr, and gustatory cortices is linked through the barrel cortex.

Taking data in *Figure 6* together, we suggest that the barrel cortex as a core station mediates the joint storage and the reciprocal retrieval of associated signals directly including the whisker signal with olfactory, gustatory, and somatic signals as well as indirectly among olfactory, somatic, and gustatory signals. The roles of barrel cortical neurons recruited as associative memory cells in the interconnection core (a wire-hub) are further strengthened by morphological and functional studies below, in addition to those data in *Figures 2–4*.

*Figure 7* illustrates the morphological studies about the suppression of convergent synapse innervations on barrel cortical neurons by neuroligin-3 knockdown. AAV-CMV-fluorescents were microinjected into the S1-Tr cortex (tdTomato in *Figure 7A*) and the piriform cortex (EBFP in *Figure 7A*). AAV-D/J8-U6-mNlgn3-EGFP was microinjected into the barrel cortex (*Figure 7A*). Neuroligin-3 knockdown in the barrel cortex appears to prevent the learning-induced formation of synapse contacts on barrel cortical neurons in neuroligin-3 shRNA mice, compared to those in scramble control mice (*Figure 7B*). The densities of synapse contacts (contacts per 100 µm dendrite) from the S1-Tr cortex are 2.25±0.33 in scramble control subgroup (red bar in left columns of *Figure 7C*, n=20 dendrites from 5 mice) and 0.35±0.13 in neuroligin-3 knockdown subgroup (blue bar, n=20 dendrites from 5 mice; p<0.01, one-way ANOVA). Synapse contacts per 100 µm dendrite from the piriform cortex are 2.50±0.19 in scramble control subgroup (red bar in right columns of *Figure 7C*, n=20 dendrites from 5 mice) and 0.30±0.15 in neuroligin-3 knockdown subgroup (blue bar, n=20 dendrites from 5 mice; p<0.01, one-way ANOVA).

In addition, *Figure 7D* demonstrates that AAV2/8-CMV-tdTomato was microinjected into the S1-Tr cortex, AAV2/8-CMV-EBFP was into the gustatory cortex, and AAV-D/J8-U6-mNlgn3-EGFP was into the barrel cortex. Neuroligin-3 knockdown appears to prevent the learning-induced formation of synapse contacts on barrel cortical neurons in neuroligin-3 shRNA mice, compared to those in scramble controls (*Figure 7E*). Synapse contacts per 100 µm dendrite from the S1-Tr cortex are 2.35±0.38 in scramble control subgroup (red bar in the left columns of *Figure 7F*, n=20 dendrites from 5 mice) and 0.28±0.08 in neuroligin-3 knockdown subgroup (blue bar, n=20 dendrites from 5 mice; p<0.01, one-way ANOVA). Synapse contacts per 100 µm dendrite from the gustatory cortex are 1.50±0.14 in scramble control subgroup (red bar in right columns of *Figure 7F*, n=20 dendrites from 5 mice) and 0.25±0.11 in neuroligin-3 knockdown subgroup (blue bar, n=20 dendrites from 5 mice; p<0.01, one-way ANOVA).

Moreover, *Figure 7G* shows that AAV2/8-CMV-tdTomato was microinjected in the piriform cortex, AAV2/8-CMV-EBFP was into the gustatory cortex, and AAV-D/J8-U6-mNlgn3-EGFP was into the barrel cortex (*Figure 7G*). Neuroligin-3 knockdown appears to prevent the learning-induced formation of synapse contacts on barrel cortical neurons in neuroligin-3 shRNA mice, compared with those in scramble control mice (*Figure 7H*). Synapse contacts per 100 µm dendrite from the piriform cortex are 2.60±0.22 in scramble control subgroup (red bar in left columns of *Figure 7I*, n=20 dendrites from 5 mice) and 0.25±0.14 in neuroligin-3 knockdown subgroup (blue bar, n=20 dendrites from

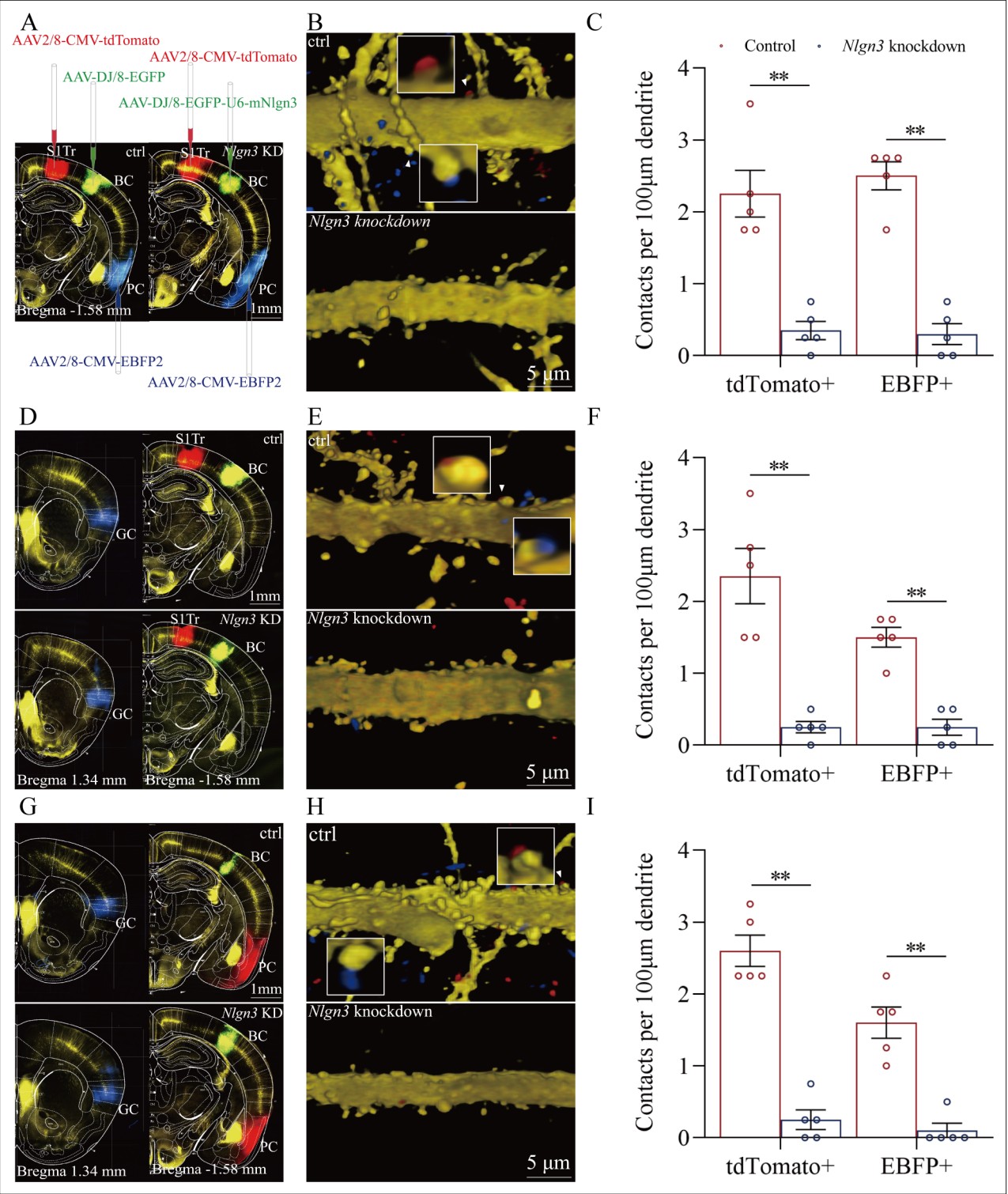

**Figure 7.** The suppression of convergent synapse innervations on barrel cortical neurons by neuroligin-3 knockdown. (**A, D, G**) AAV-DJ/8-EGFP was injected into the BC in scramble control mice and AAV-DJ/8-EGFP-U6-m*Nlgn3* was injected into the BC in neuroligin-3 short-hairpin RNA (shRNA) mice before training. AAV2/8-CMV-tdTomato was microinjected into the S1-Tr cortex, AAV2/8-CMV-EBFP was into the piriform cortex (A). AAV2/8-CMV-tdTomato was microinjected into the S1-Tr cortex, AAV2/8-CMV-EBFP was into the gustatory cortex (D). AAV2/8-CMV-tdTomato was microinjected into the piriform cortex, AAV2/8-CMV-EBFP was into the gustatory cortex (G). (**B, E, H**) Neuroligin-3 knockdown appears to prevent the learning-induced formation of the synapse contacts on barrel cortical neurons in neuroligin-3 shRNA mice, compared to those in scramble control mice. (**C, F, I**) The densities of synapse contacts (contacts per 100 μm dendrite) between the spines of yellow fluorescent protein (YFP)-labeled neurons and the axon boutons labeled by EBFP or tdTomato in scramble control group (red bar) compared to those in neuroligin-3 knockdown group (blue bar).

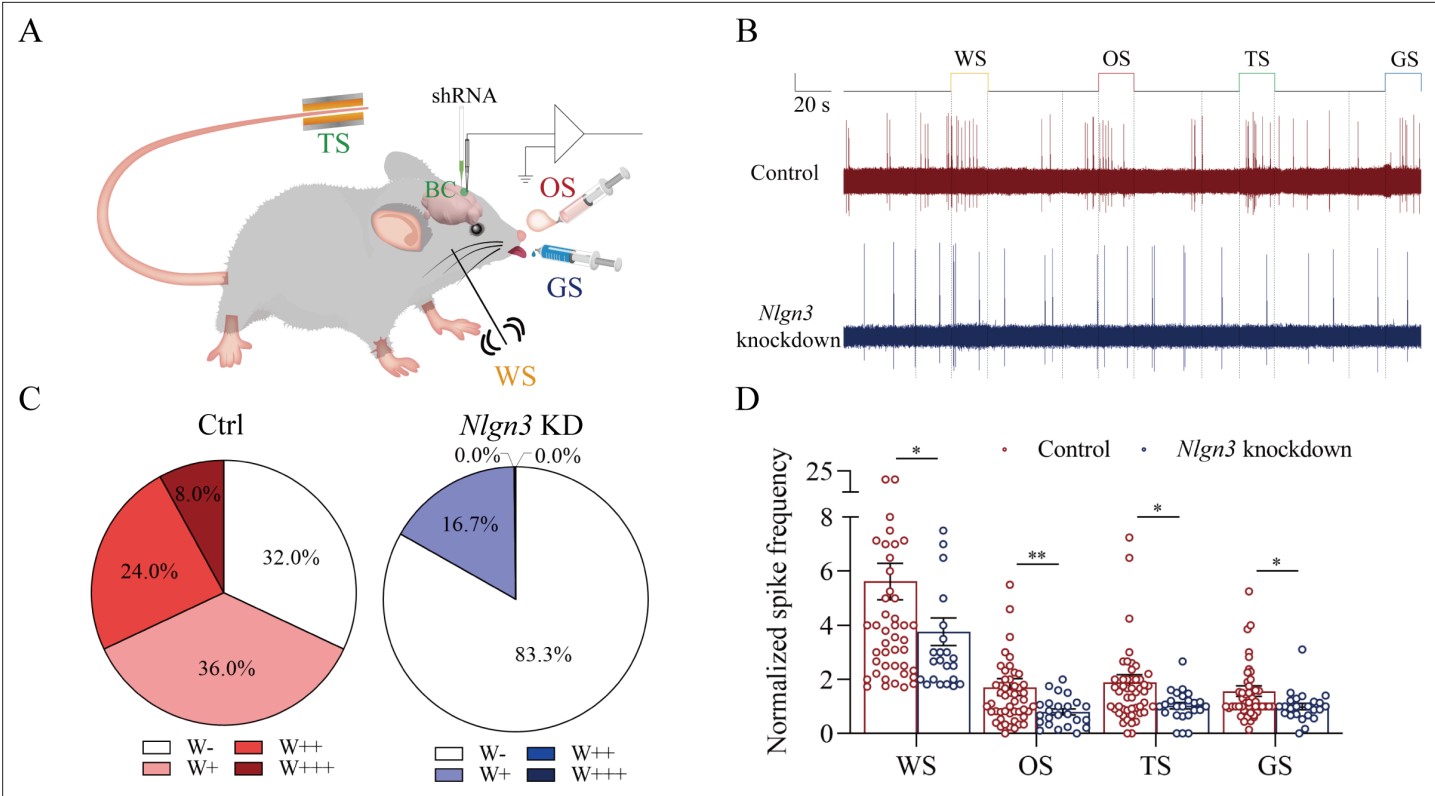

**Figure 8.** The response of barrel cortical neurons to odor, tail-heating, and gustatory sucrose signals alongside the whisker tactile signal were downregulated in sh*Nlgn3* mice. (**A**) A diagram illustrates microinjection of adeno-associated viruses (AAVs) and a recording of local field potentials (LFPs) in the barrel cortex. (**B**) An example of barrel cortical neuron in response to butyl acetate, tail-heating, sucrose, and whisker tactile signals from scramble mouse (red trace) and sh*Nlgn3* mouse (blue trace). The calibration bars are 2 mV and 20 s. (**C**) The percentages of associative neurons in scramble mice (left panel) and in sh*Nlgn3* mice (right panel). Neurons that respond only to whisker stimulus (WS) are labeled as W-, and those that respond to WS and one, two, or all three other signals are labeled as W+, W++, and W+++, respectively. (**D**) The normalized spike frequencies in response to whisker tactile signals, butyl acetate, tail-heating signal, and sucrose signal in scramble mice (red bar) and in sh*Nlgn3* mice (blue bar), respectively.

5 mice; p<0.01, one-way ANOVA). Synapse contacts per 100 µm dendrite from the gustatory cortex are 1.60±0.22 in scramble control subgroup (red bar in the right columns of *Figure 7I*, n=20 dendrites from 5 mice) and 0.10±0.1 in neuroligin-3 knockdown subgroup (blue bar, n=20 dendrites from 5 mice; p<0.01, one-way ANOVA). The suppression of the new synapse contacts on barrel cortical neurons inputted from piriform, S1-Tr, and gustatory cortices by neuroligin-3 knockdown indicates that the formation of new synapses and the recruitment of associative memory neurons require neuroligin-3-mediated synapse linkage. It is noteworthy that the learning-induced connection from the hippocampus to the barrel cortex disappears by the knockdown of barrel cortical neuroligin-3, indicating that new synapse innervations from the hippocampus to other cortices of memorizing specific signals is essential for the hippocampus to strengthen specific memory in cerebral cortices.

*Figure 8* presents the functional study about the suppression of associative memory neuron recruitment in the barrel cortex by the neuroligin-3 knockdown. The responses of barrel cortical neurons to butyl acetate, tail-heating, or sucrose signals alongside the whisker tactile signal were monitored by recording their electrophysiological activities induced by these signals (*Figure 8A*). If the recruitment of barrel cortical neurons to be associative memory neurons are suppressed by neuroligin-3 knockdown, the number and the activity strength of associative memory neurons should be lower in the neuroligin-3 shRNA subgroup than in the scramble control subgroup (*Feng et al., 2017*; *Wang et al., 2019b*; *Wu et al., 2020*). *Figure 8B* shows examples of barrel cortical neurons in response to those butyl acetate, tail-heating, sucrose and whisker tactile signals from scramble control subgroup (red trace) and neuroligin-3 shRAN subgroup (blue trace), respectively. Barrel cortical neurons in a scramble control mouse appear to respond to butyl acetate, sucrose, and tail-heating signals alongside

the whisker tactile signal, in comparison with the barrel cortical neurons from a neuroligin-3 shRNA mouse. Percentages of barrel cortical neurons in response to the whisker signal plus butyl acetate, tail-heating, and/or sucrose signal are 68% in the scramble control subgroup (n=50 neurons from 5 mice) and 16.7% in in neuroligin-3 knockdown subgroup (n=24 neurons from 5 mice, p<0.01, $\chi^2$-test in *Figure 8C*). Based on the analysis of neuronal spike frequency in response to these signals, or activity strength, the normalized spike frequencies (Hz) in response to the whisker tactile, butyl acetate, tail-heating, and sucrose signals are 5.62±0.67, 1.71±0.31, 1.89±0.28, and 1.56±0.19 in scramble control subgroup (red bars in *Figure 8D*; n=50 neurons from 5 mice) as well as 3.76±0.51, 0.8±0.11, 1.02±0.12, and 1.00±0.12 in neuroligin-3 knockdown subgroup (blue bars, n=24 from 5 mice; p<0.05, one-way ANOVA). Thus, the learning-induced recruitment of associative memory neurons that encode multi-modal signals including the whisker, olfactory, tail, and gustatory signals in the barrel cortex requires the synapse linkage mediated by neuroligin-3.

## Discussion

In our studies, mice receive a few pairs of cross-modal signals sequentially including whisker tactile plus olfaction signals, whisker tactile plus tail signals, and whisker tactile plus gustatory signals. This associative learning by multiple pairs of cross-modal signals leads to the reciprocal form of associative memory, in which the reciprocal retrievals of these associated signals include odorant-induced whisker motion with whisking-induced olfactory response, tail-heating-induced whisker motion with whisking-induced tail swing, and gustation-induced whisker motion with whisking-induced gustatory response. In this model of associative learning, the whisker tactile signal is a common signal in the pairs of associated signals. After memory formation, the whisker motion is the common response induced by the paired signals during the retrieval of associative memory (*Figure 1*). The whisker tactile sensation as a core signal commonly shared for olfactory, gustatory, and tail signals is based on mutual synapse innervations between the neurons in the barrel cortex and those neurons in the piriform, S1-Tr, and gustatory cortices (*Figure 2*). Moreover, certain barrel cortical neurons receive convergent synapse innervations from piriform, S1-Tr, and gustatory cortices (*Figure 3*) as well as encode the signals from these sensory cortices (*Figure 4*). Such morphological and functional data indicate that barrel cortical neurons have been recruited as associative memory cells to encode associative memory correlated to the reciprocal retrievals of these associated signals in behavior tasks. Interestingly, with the interconnections of the barrel cortex with piriform, S1-Tr, and gustatory cortices, the excitation of piriform cortical neurons by the olfactory signal may indirectly activate S1-Tr and gustatory cortical neurons intermediated by the active barrel cortex, such that the odorant-induced tail swing and odorant-induced gustatory response emerge in the retrieval of associative memory, or the other way around (*Figure 5*). If the reciprocal retrievals of the associated signals between the whisker signal as a common signal and olfactory, tail and gustatory signals are thought of as the first order of associative memory (*Wang, 2019a*), the reciprocal retrievals of the non-associated signals including olfactory, tail, and gustatory signals would be the secondary order of associative memory, upgrading those terms of the first order and the secondary order of Pavlov's conditioning (*Wasserman and Miller, 1997*). Our studies about the influence of knocking neuroligin-3 (one of linkage proteins for synapse fixation) on associative memory and associative memory neurons (*Figures 6–8*) indicate that neuroligin-3 has been one of molecular substrates to participate in the formation of associative memory and the recruitment of associative memory neurons.

Associative learning and memory are often investigated by the animal model of conditioned reflexes, such as classical conditioning and operant conditioning (*Byrne, 1987*; *Wasserman and Miller, 1997*). The classical conditioning usually includes Pavlov's salivary-secretion conditioning, eye-blinking conditioning, and body-freezing conditioning induced by the bell ring (*Bracha et al., 2009*; *Burhans et al., 2008*; *Davis et al., 1993*; *Maren, 2008*; *Perkowski and Murphy, 2011*; *Reijmers et al., 2007*; *Woodruff-Pak and Disterhoft, 2008*). The operant conditioning includes the pedal-pushing and the conditioned place preference induced by the foods or drugs (*Aguilar et al., 2009*; *Bardo and Bevins, 2000*; *DeLong, 1971*; *DeLong, 1972*; *Prus et al., 2009*; *Tzschentke, 1998*; *DeLong, 1973*; *McKendrick and Graziane, 2020*). All of these conditioning models were formed by the associations of the bell ring with food, air-puffing, or electrical shock in the classical conditioning as well as the light with the pedal-pushing for food or drug rewards in the operant conditioning. In the studies of memory retrievals, the bell ring or the light was used to induce the conditioned responses

that were evoked originally by the innate signals. Beyond these conditionings with a single direction, associative memory formed by the association of two or more signals in lifespan can be induced reciprocally. One signal evokes the responses originally induced by other signals, or the other way around, for example, the reciprocal retrievals between words in the sound signal and words-presented images in the visual signal. The reciprocal form of the associative memory has been previously shown in the associations of the whisker signal and the olfactory signal (*Feng et al., 2017*; *Wang et al., 2015*; *Gao et al., 2019*; *Yan et al., 2016*). The reciprocal form of associative memory through the associations of the whisker signal with the tail signal and of the whisker signal with the gustatory signal is presented in *Figure 1*. In addition, the reports in the previous conditioning models have not systemically denoted the relationship between the common core signal (the bell ring or the light) and their paired signals. *Figure 5* in the present study shows that the whisker signal is the core signal commonly shared by multiple signals for their direct and indirect retrievals in the reciprocal form of the associative memory. In terms of their cellular substrates, our studies reveal the formation of new synapse interconnections between coactivated cerebral cortices by the barrel cortex as the core station or the hub (*Figure 2*). These data have not been thought and examined in previous studies about various conditioning models.

The accumulation and the enrichment of specific memory contents in the brain are fulfilled by learning more and more pairs of associated signals during postnatal development. One signal among these pair signals may become the common signal shared for their reciprocal retrievals and the cognitions (*Wang, 2019a*; *Schacter et al., 2020*). The apple's auditory signal can be a common signal shared by the visual signal of its shape and color, the gustatory signal of its taste and the olfactory signal of its smell. The listening of this apple's vocal sound can induce the reciprocal retrievals of its other cross-modal features. The thunderstorm often associates heavy rain, something wetness and even flooding. The thunderstorm can lead to the reasoning about the forthcoming of heavy rain and flooding in the cognitive activities as well as the worry about their happening in the emotional reactions. To elucidate the cellular and molecular substrates for these cognitive and emotional activities commonly seen in the lifespan, we expect to establish an animal model in consistence with these phenomena. In the present study, we develop a mouse model of associative learning by giving multiple pairs of associated signals in distinct time points, in which one of such paired signals is a common signal for others. The associations of the whisker tactile with the olfactory signal, the tail-heating signal, and the gustatory signal are sequentially given to the mice (*Figure 1*). After this associative learning, the reciprocal forms of the associative memory emerge in these mice, including odorant-induced whisker motion with whisking-induced olfactory response, tail-heating-induced whisker motion with whisking-induced tail swing, and gustation-induced whisker motion with whisking-induced gustatory response. The whisker signal is set as the common core of these signals and the whisker fluctuation appears to become the common core of their responses (*Figure 1*). Therefore, the animal model with one signal to be a common core signal for the reciprocal form of the associative memory and the logical reasoning is established in our studies.

In terms of cellular substrates underlying these memory retrievals, cognitions and emotions with the common signal, the neuronal assemblies in the cerebral cortex for one modality may interconnect those neuronal assemblies in the cerebral cortices for other modalities (*Feng et al., 2017*; *Wang, 2019a*), and these neuronal assemblies may encode all of these associated signals (*Feng et al., 2017*; *Vincis and Fontanini, 2016*; *Wang et al., 2015*; *Wang et al., 2013*; *Wang, 2019a*). In other words, associative memory neurons are recruited to encode multi-modality signals after associative learning (*Wang, 2019a*). This hypothesis has been examined and proved in our studies. With experiencing the associative learning, mutual axon projections and synapse interconnections emerge between the neurons in the barrel cortex and the neurons in piriform, S1-Tr, and gustatory cortices (*Figure 2*). Barrel cortical neurons receive the convergent synapse innervations from piriform, S1-Tr, and gustatory cortices (*Figure 3*). Barrel cortical neurons become able to encode the signals inputted from these sensory cortices (*Figure 4*). The synapse interconnections formed among the coactive cortices and the associative memory neurons recruited in the cortical area to encode a common core signal plus other associated signals endorse the hypothesis that memory retrievals, cognitive activities, and emotional responses are based on the common signals memorized (*Wang, 2019a*). Morphological and functional data in this study suggest that the barrel cortical neurons have been recruited as associative memory neurons to encode the joint storages and reciprocal retrievals of the whisker signal

with other associated signals in behavioral tasks. Based on these associative memory neurons to encode a common signal plus other signals associatively learned previously, any one of these signals is able to induce the retrievals of other signals in associative memory as well as the associative processes during cognitive activities and emotional responses (*Wang, 2019a*).

The associated signals in memories are often retrieved in any time of the lifespan. Based on the rules, the coactivity together and the interconnection together (*Wang, 2019a*) as well as coactivity together and the strengthening together (*Hebb, 1949*), those cortical areas are interconnected and connection-strengthened after the learning of associated signals and the memory to these signals. With the repetitive activations of these interconnected neurons for the retrievals of those associated signals, associative memory neurons in the single modality cortex become a core station to translate those signals encoded by other cortices as well as to forward the signals from one of its interconnected cortices to others of its interconnected cortices, or the other way around (*Figures 3–4*). With these interconnections and associative memory neurons in the core station (or the hub), any one of these signals surrounding the signal encoded in the hub may become the signal associated with other surrounding signals, which have not been directly associated in previous associative learnings. Our studies in *Figures 5–6* show this type of indirect retrievals of non-associated signals, likely they seem associated. The olfactory signal induces the retrievals of tail and gustatory signals, or the other way around, indirectly mediated by the active barrel cortex. This datum also explains those phenomena in the retrievals of non-associatively memorized signals in a cross-core manner indirectly. For instance, the apple's shape signal in the mind can retrieve its taste signal or odor signal over the apple sound signal. The heavy rain signal retrieves the flooding signal over the thunderstorm signal in the mind. The data in behavior tasks and cellular mechanisms underlying these first-order and secondary-order associative memories in the retrieval of many signals are critically important to reveal a wide range of brain functions and their neural substrates.

Molecular substrates for the formation of new axon projections and synapse innervations as well as the recruitment of associative memory neurons in cerebral cortices have to be addressed. The microtubule prolongation and the synapse formation require activity-dependent epigenetic events from microRNA-324 and microRNA-133a (*Feng et al., 2017*; *Lei et al., 2017*; *Wu et al., 2020*). As we know, the formation of new synapses is followed by the linkage and fixation of presynaptic and postsynaptic membranes through linkage proteins, such as neuroligin, neurexin, N-cadherin, ephrin, and so on (*Craig and Kang, 2007*; *Li et al., 2022*; *Lisé and El-Husseini, 2006*; *Südhof, 2017*; *Uchigashima et al., 2021*). In the present study, we have studied the requirement of neuroligin-3 for the synapse linkage and fixation. Neuroligin-3 mRNA knockdown by its specific shRNA in the barrel cortex significantly inhibits the formation of new synapses innervated from coactive cortices and the recruitment of associative memory neurons in the barrel cortex (*Figures 7–8*), in addition to the suppression of associative memory formation (*Figures 5–6*). Therefore, our studies indicate that synapse linkage and fixation are essential for the recruitment of associative memory neurons based on the formation of new synapses. With the rule of coactivity together and interconnection together (*Wang, 2019a*), the cascade from the inputs of associated signals during associative learning to the recruitment of associative memory neurons for memory formation may include these following steps, the coactivity of somatosensory cortical neurons by their intense action potentials, the alternation of epigenetic processes by microRNA-324/133, the expression of neuroligin-3, and the synapse linkage for new synapse formation in these cortical neurons. In addition to neuroligin-3, other molecules including NET3 and ttbk1 have been found to be involved in the formation of new synapse innervations and the recruitment of associative memory neurons for the formation of associative memory (*Feng et al., 2017*; *Lei et al., 2017*; *Wu et al., 2020*; *Wang et al., 2019b*).

Associative memory neurons are recruited in the formation of associative memory after the associative learning by pairing whisker tactile, olfaction, gustation, and tail temperature signals in the present study and others (*Feng et al., 2017*; *Wang et al., 2015*; *Wang, 2019a*; *Lei et al., 2017*; *Wu et al., 2020*). Associative memory neurons are featured by their recruitment from coactive neurons, the synapse interconnections among coactive neurons, the convergent synapse innervations onto these coactive neurons from the new and innate inputs, as well as the encoding of multiple signals inputted from these synapse innervations on these coactive neurons (*Wang, 2019a*). In another study, the social stress by the association of the stressful signals, including the battle sound produced from the mouse fighting, the pain from body injury regions, and the visual image from the fighting scene,

results in fear memory and anxiety. Some neurons in the somatosensory cortex interconnect with auditory and visual cortical neurons receive the synapse innervations from the auditory cortex, the visual cortex, and the thalamus, as well as encode these stressful signals from auditory, visual, and somatosensory systems, that is, recruited as primary associative memory neurons. Moreover, the prefrontal cortical neurons are recruited as secondary associative memory neurons by receiving convergent synapse innervations from these sensory cortices as well as encode their inputted stressful signals (our data ready to be published). Two lines of evidences reveal the recruitment of associative memory neurons during associative learning. With such two mouse models of the associative learning, we have identified associative memory neurons for the joint storage and the reciprocal retrieval of multiple signals associated during the learning, which strengthens the concept of associative memory cells as basic units in memory traces or engrams for the associative learning and memory (*Wang, 2019a*). This strengthened conclusion also encourages those memoriologists to examine whether associative memory neurons are recruited in other types of the associative learning and memory under their investigations.

Associative memory is based on the recruitment of associative memory neurons. Moreover, the function and the connection of these neurons are strengthened. For instance, barrel cortical neurons are functionally upregulated in their spiking frequencies in response to the input signals (*Figures 4 and 8*) as well as their morphological interconnections with other brain areas are raised (*Figure 2*). The decreases in spiking thresholds and refractory periods as well as the increase of synapse-driving force strengthen neuronal activities. The morphological upregulation of synapse interconnections among associative memory neurons can enhance their spike-encoding ability. In addition, the coactivity of associative memory neurons may trigger the activation of intracellular Ca2+/calmodulin signaling pathway. This signaling pathway can initiate the conversion of silent synapses into functional synapses and the conversion of inactive synapses into active synapses to strengthen the synapse-driving force and increase the activities of associative memory cells (*Wang, 2019a*; *Liao et al., 1995*; *Wang and*

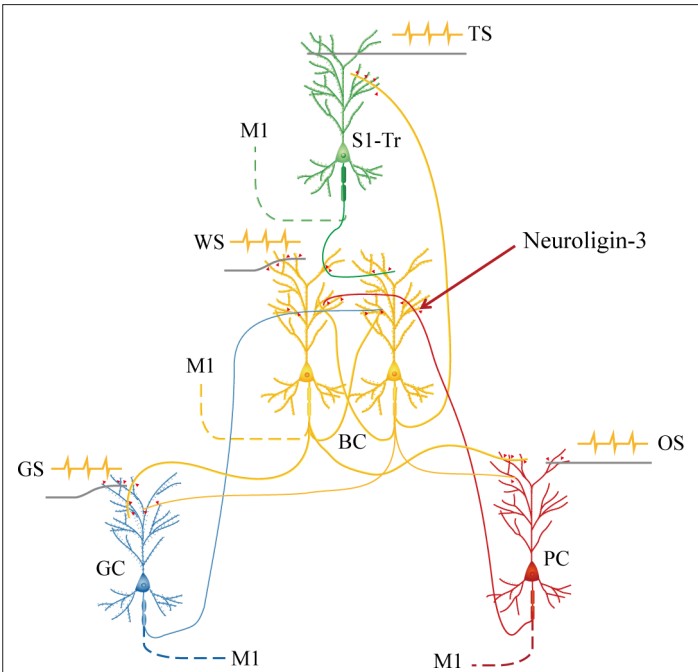

**Figure 9.** Barrel cortical neurons are recruited to be associative memory neurons in the hub-like core station, based on the synapse interconnections with piriform, S1-Tr, and gustatory cortices, in addition to their intramodal interconnection. Molecular substrates for these cellular changes are based on the activity-dependent epigenetic processes and neuroligin-3-mediated synapse linkage. The associative learning by pairing the whisker stimulus (WS) with the odor stimulus (OS), the WS with the tail-heating stimulus (TS), and the WS with gustatory stimulus (GS) induces mutual synapse innervations from PC (red), S1-Tr (green), and GC (blue) to BC (yellow), as well as from BC to PC, S1-Tr, and GC, which drives the recruitment of barrel cortical neurons to be associative memory cells. The newly formed neural circuits constitute the foundation of multiple cross-modal memories.

*Kelly, 2001*) as well as to upregulate neuron-encoding ability (*Zhang et al., 2004*; *Chen et al., 2008*), which leads to the strengthening of associative memory. Taken these together, our studies support the hypothesis about the activity-dependent positive recycle in the recruitment and the refinement of associative memory neurons for the formation and the strengthening of associative memory (*Wang, 2019a*). In other words, our studies endorse two ideas in the field of memorioscience, that is, the coactivity together and the interconnection together (*Wang, 2019a*) as well as the coactivity together and the strengthening together (*Hebb, 1949*). Memorioscience refers to science in relevance to discovering the regulations and mechanisms of memory formation and strengthening under physiological and pathological conditions as well as to developing the approaches for the improvement of memory deficits.

In summary, our studies present that the barrel cortex can become the core station for the first-order and secondary-order associative memory. Barrel cortical neurons can be recruited as associative memory neurons, in addition to the interconnections between the barrel cortex and piriform, S1-Tr, and gustatory cortices. Animals are able to conduct the retrievals of newly learned signals based on these new synapse interconnections in the barrel cortex as the wire-hub among coactive cortices, alongside the retrieval of the innate signals through their innate circuits (*Figure 9*). Molecular substrates for such cellular changes are based on the activity-dependent epigenetic processes and neuroligin-3-mediated synapse linkage. Thus, the coactivity of the cerebral cortices in the associative learning induces the formation of their interconnection, which can endorse the first order and the secondary order of associative memory. Associative memory neurons in the cerebral cortices are recruited by mutual synapse innervations based on neuroligin-3-mediated synapse linkage. Our study appears an initiative that reveals the associative memory neurons and their assemble circuits as the neural basis of cognitive activities and emotional responses in the field of memory science. In this regard, the learning of associated signals can be thought of as the coactivation of sensory cortices that encode these signals. Associative memory can be thought of as the activity-dependent formation of new interconnections among these cortices and the convergent synapse innervations on coactive neurons by new synapses and previous synapses, which is the feature of the recruitment of associative memory neurons in the early development to memorize unitary and simple signals, as well as the activity-dependent strengthening of these synapse connections among associative memory neurons in the following periods to memorize those complex signals organized from unitary/simple signals. The retrieval of associative memory can be thought of as the reactivation of these associative memory neurons by exogenous cues or endogenous thinking (*Wang et al., 2019b*).

## Materials and methods
### Studies approved in mice
Experiments were conducted in accordance with guidelines and regulations by the Administration Office of Laboratory Animals in Beijing China. All experimental protocols were approved by the Institutional Animal Care and Use Committee in Administration Office of Laboratory Animals in Beijing China (B10831). C57BL/6JThy1-YFP mice (Jackson Lab. JAX stock #003782, USA) were applied in our studies, whose glutamatergic neurons in the cerebral brain were genetically labeled by YFP (*Feng et al., 2000*; *Zhang et al., 2013*). The mice were accommodated in specific pathogen-free facilities with a circadian of 12 hr for day and night plus the sufficient availability of the food and water. The mice in postnatal 20 days and well-developed bodies were selected for starting the training of associative learning. These mice were taken into the laboratory for them to be familiar with experimental operators and the training apparatus for 2 days. The timeline of our experiments was executed in the following steps, the separation of these mice randomly into PSG and UPSG, the injections of AAVs into their barrel cortices, the uses of the training paradigm in mice for 2 weeks after this surgical operation of AAV injection was done about 48 hr, the maintenance of these mice in their living houses about 5 days, and then the morphological and electrophysiological studies.

### Behavioral study
The paradigm of associative learning was pairing the signals of whisker tactile with odorant, gustation, or tail-heating in mice (*Feng et al., 2017*; *Vincis and Fontanini, 2016*). The whisker signal was mechanical vibrate stimuli (5 Hz) to longer whiskers for 20 s, which were the contralateral side of the

barrel cortex for searching associative memory neurons recruited during associative learning. The whisker signal intensity was enough to evoke whisker fluctuations, or the innate whisking-induced whisker motion. The odor signal was the butyl acetate pulse closely to the noses for 20 s, which was given by switching on a butyl acetate-containing tube to generate a small liquid drop in front of the noses. The intensity of butyl acetate reached to the level of activating olfactory bulb neurons, which had been ensured by the two-photon cell imaging (*Wang et al., 2015*). The tail-heating signal was given by using the heat plate that touch the distal ends of mouse tail. The tail-heating signal intensity was about 45 ± 2°C that was enough to induce the tail swing away from this heating plate within 10 s. This temperature did not make the injury of thick skin on the tail (*Feng et al., 2000*). The gustatory signal was administered by giving a sucrose drop (0.06 mol/l) to mouse lips and teeth, where this concentration sufficiently induced tongue lickings to mouth lips (*Vincis and Fontanini, 2016*). The response to this sucrose in the mice was the licking of their tongues to their mouth lips within 20 s.

The detailed paradigm of the associative learning has been described in our previous studies (*Wang et al., 2015*; *Wang, 2019a*; *Liu et al., 2017*; *Yan et al., 2016*). Briefly, the mice were divided into two groups, PSG and UPSG. In PSG mice, the paradigms of the associative learning were the pairing of whisker tactile with butyl acetate stimulations for 20 s, the pairing of whisker tactile with tail-heating for 20 s, and the pairing of whisker tactile with sucrose for 20 s sequentially, in which the intervals for each of these pairing stimulations were 5 min. The paired-stimulations to PSG mice in the paradigm of associative learning were lasted for 20 s within each of times, were given five times per day with 2 hr intervals, and were used for 12 days. In UPSG mice, the paradigm for unpaired-stimulations was featured by whisker tactile, butyric acid, tail-heating, and sucrose stimulations sequentially, where the intervals for each of these stimulations were 5 min. The unpaired-stimulations to all UPSG mice lasted for 20 s in each of individual stimulations and five times per day in 2 hr intervals for 12 days. The setting of these paradigms was based on a fact that the onsets of reciprocal retrievals in these associated signals reached to their plateau level by training approximately 10 days (*Wang et al., 2015*; *Liu et al., 2017*). The intensity, duration, and frequency of these stimulations were digitally set by a multiple sensory modal stimulator with the locked parameters for all mice.

Whisker fluctuations evoked by odorant (butyl acetate), gustation (sucrose), and tail-heating were measured to identify the retrieval of associative memory that had been formed during the paired-training. On the other hand, the olfactory response to butyl acetate by odorant selection in T-maze, the gustatory response to sucrose by tongue-licking lips, as well as the pain response to tail-heating by tail swing, which were induced by the whisker stimulation, were measured to identify the reciprocal retrieval of associative memory that had been formed in paired-training (*Feng et al., 2017*; ; *Gao et al., 2016*; *Gao et al., 2019*; *Lei et al., 2017*; *Wu et al., 2020*). To quantify the onset time and the strength of whisker responses induced by odorant, gustation, or tail-heating, the whisker fluctuations in response to these testing stimulations (20 s) of butyl acetate, sucrose, and tail-heating were recorded by the digital video camera (HDR-AS100V, SONY, Japan; 240 fps) after the training. The whisker motions induced by butyl acetate, sucrose, or tail-heating, or the formation of associative memory, were accepted if whisker fluctuations met the following criteria. The patterns of odorant-induced whisker motion, gustation-induced whisker motion, and tail-heating-induced whisker motion were similar to innate whisker motions induced by the whisker stimulation, but differed from spontaneous low-magnitude whisking. The whisking frequencies and angles raised significantly in PSG mice, compared to those in baseline controls and UPSG mice. The patterns of whisker fluctuations in odorant-induced whisker motion, gustation-induced whisker motion, and tail-heating-induced whisker motion were originally presented in the whisker stimulus, such that the odorant, gustatory, and tail signals triggered the recall of the whisker signal and then the whisker motions similar to the innate reflex (*Wang et al., 2015*; *Wang, 2019a*; *Liu et al., 2017*). In addition, this camera was also used to take the videos for monitoring the tail-swing and lip-licking in these mice to quantify the latency, the duration, the times, and the frequency of these processes.

## Neural tracing to search associative memory cells morphologically

Strategical approaches to examine how basic neuronal units in memory traces interconnected and interacted each other included the morphological identification of their synapse connections by applying AAV-carried fluorescent proteins as well as the electrophysiological recording of associative memory neurons in response to multiple associated signals in the mice that had experienced

associative learning and expressed associative memory (*Feng et al., 2017*; *Gao et al., 2016*; *Gao et al., 2019*; *Lei et al., 2017*; *Wu et al., 2020*). A few of CMV-coded AAVs were used in our experiments, such as AAV2/8-CMV-EGFP, -EBFP, and -tdTomato as well as AAV2/retro-CMV-EGFP (OBiO Inc, Shanghai, China). In the study of the morphological interconnections of the barrel cortex with the piriform cortex, the gustatory cortex or the S1-Tr cortex, AAV2/8-CMV-tdtomato, and AAV/retro-CMV-EGFP were microinjected into the barrel cortex (–1.5 mm posterior to the bregma, 3 mm lateral to the middle line, and 0.7 mm depth away from the cortical surface; *Paxinos and Watson, 2005*) before the training paradigm. The microinjection of AAVs into cerebral cortices was conducted by using a glass electrode. The injection quantity and duration were controlled by a microsyringe system held with the three-dimensional stereotaxic apparatus (RWD Life Science, Shenzhen, China). The quantities of microinjected AAVs were 0.5 µl for AAV2/retro-CMV-EGFP and 0.2 µl for AAV2/8-CMV-tdTomato with an injection period about 30 min. Theoretically and practically, AAV2/8-CMV-tdTomato was uptaken and then expressed in barrel cortical neurons, where the red fluorescent protein (RFP) was produced. The RFP, or tdTomato, was transported toward entire axons of such barrel cortical neurons at their target areas in an anterograde manner, such that axonal boutons and terminals were labeled by the RFP (*Feng et al., 2017*; *Gao et al., 2016*; *Gao et al., 2019*; *Lei et al., 2017*; *Wu et al., 2020*). AAV/retro-CMV-EGFP was uptaken by axonal terminals and boutons, and then was transported to the somata of cortical neurons in the retrograde manner for its expression and production, so that the sources of neuronal somata whose axons projected to the barrel cortex were tracked (*Tervo et al., 2016*).

In the study of convergent synapse innervations on barrel cortical neurons from the piriform cortex, gustatory cortex, and S1-Tr cortex, that is, associative memory neurons in the barrel cortex, CMV-coded AAVs were injected into the piriform cortex (AAV2/8-CMV-EGFP 0.2 µl; –0.6 mm posterior to the bregma, 3.7 mm lateral to the middle line, and 3.8 mm depth below the cortical surface), the gustatory cortex (AAV2/8-CMV-EBFP 0.2 µl; 1.1 mm posterior to the bregma, 3.1 mm lateral to the middle line, and 1.9 mm depth below the cortical surface) and the S1-Tr cortex (AAV-CMV-tdTomato 0.2 µl; –1.5 mm posterior to the bregma, 1.5 mm lateral to the middle line, and 0.5 mm depth away from the cortical surface), respectively. These CMV-coded AAVs were uptaken and expressed at cortical neurons within their injected areas, where these fluorescent proteins were produced. These fluorescent proteins were then transported toward entire axons at target areas in an anterograde manner, so that their axon boutons and terminals labeled by such fluorescent proteins were detected in target areas and even onto dendritic spines of barrel cortical neurons, that is, the morphological identification of associative memory neurons (*Feng et al., 2017*; *Gao et al., 2016*; *Gao et al., 2019*; *Lei et al., 2017*; *Wu et al., 2020*). The microinjections of CMV-coded AAVs were done 2 days before the training paradigm to allow the transportation of expressed fluorescent proteins to entire axon boutons and terminals along with the projection of learning-induced axon growth.

After the injections of AAVs in PSG and UPSG mice were about 3 weeks when the training paradigm has been done, the mice were anesthetized by intraperitoneal injections of 4% chloral hydrate (0.1 ml/10 g) and perfused through the left ventricle with 50 ml 0.01 M phosphate buffer solution (PBS) followed by 50 ml of 4% paraformaldehyde until their bodies were rigid. The brains were quickly isolated and post-fixed in 4% paraformaldehyde for additional 24 hr. The cerebral brains were sliced by a vibratome in a series of coronal sections with a thickness of 100 µm. In order to clearly show the three-dimensional images about new synapses in the barrel cortex, brain slices were placed into Sca/eA2 solution for 10 min to make them transparent (*Gao et al., 2019*; *Hama et al., 2011*). These slices were rinsed by PBS for three times, air-dried, and cover-slipped. The images of neurons, dendrites, dendritic spines, and axonal boutons were taken and collected at a ×60 lens for high magnification in a confocal microscope (Nikon A1R plus). The anatomic images of the cerebral brain were taken by a ×4 lens for low magnification in this confocal microscope. In C57BL/6J Thy1-YFP mice, postsynaptic neuron dendrites and spines were labeled by the YFP. The presynaptic axon boutons were labeled by the GFP, BFP, and RFP produced from CMV-coded AAVs being injected, respectively. The contacts between yellow dendritic spines and green, blue, or red axon boutons with less than 0.1 µm space cleft were presumably chemical synapses (*Gao et al., 2019*; *Wu et al., 2020*). The wavelength of an excitation laser beam 488 nm was used to activate the GFP and YFP. The wavelength of an excitation laser beam 561 nm was used to activate tdTomato. The wavelength of an excitation laser beam 405 nm was used to activate the BFP. The wavelengths of the emission spectra of the BFP, GFP, YFP,

and RFP were 412–482 nm, 492–512 nm, 522–552 nm, and 572–652 nm, respectively. The images of dendritic spines, axon boutons, as well as synapse contacts were analyzed quantitatively by ImageJ and Imaris (*Gao et al., 2019*). The associative memory neurons were accepted by detecting at least two sources of boutons onto the dendritic spines of YFP-labeled barrel cortical neurons (*Feng et al., 2017*; *Wang, 2019a*).

## Neuronal recordings to search associative memory cells

Before the electrophysiological recording of barrel cortical neurons, the mice in PSG or UPSG were anesthetized by intraperitoneal injections of 4% choral hydrate (0.1 ml/10 g) for surgical operations after training paradigms had been done. The body temperature was kept at 37°C by a computer-controlled heating blanket. The craniotomy (2 mm in diameter) was done on the mouse skull above the left side of the barrel cortex, or a contralateral side of whisker stimulation (–1.34 mm posterior to the bregma and 2.75 mm lateral to the midline) (*Wu et al., 2020*). Electrophysiological recordings to barrel cortical neurons in vivo were conducted in the mice under the light anesthetic condition with the withdrawal reflex by pinching, the eyelid blinking reflex by air-puffing, and the muscle relax. The unitary discharges of cortical neurons in the category of local field potential were recorded in layers II-III of the barrel cortices by using glass pipettes filled with a standard solution (150 mM NaCl, 3.5 mM KCl, and 5 mM HEPES). The resistance of those recording pipettes was 30–50 MΩ. The electrical signals of barrel cortical neurons in their spontaneous spikes and evoked spikes by the whisker, odorant, gustatory, or tail stimulations were recorded and acquired by AxoClamp-2B amplifier and Digidata 1322A, and were analyzed by pClamp 10 system (Axon Instrument Inc, CA, USA). Spiking signals were digitized at 20 kHz and filtered by a low-pass at 5 kHz. The 100–3000 Hz band-pass filter and the second-order Savitzky-Golay filter were used to isolate the spike signal. Spiking frequencies were quantitatively analyzed. Relative spike frequencies in response to the whisker, odorant, tail, and gustatory stimulations were the ratio in that spike frequencies in response to these stimuli were divided by spontaneous spike frequencies in 20 s before the stimulations. When the ratio of evoked-spike frequencies by a stimulus to spontaneous spike frequencies reached 1.7 or above, barrel cortical neurons were deemed as the response to this stimulus (*Feng et al., 2017*; *Gao et al., 2016*; *Gao et al., 2019*; *Wu et al., 2020*). The associative memory neurons were accepted by detecting a situation that barrel cortical neurons respond to at least two sources of those paired signals (*Feng et al., 2017*; *Wang, 2019a*).

## The study of molecular mechanism for the recruitment of associative memory cells

In the study of the role of neuroligin-3, one of the proteins for synapse linkage (*Craig and Kang, 2007*; *Li et al., 2022*; *Lisé and El-Husseini, 2006*; *Südhof, 2017*; *Uchigashima et al., 2021*), in the formation of new synapse innervations and the recruitment of associative memory cells in the barrel cortex, the approach of mRNA knockdown was applied by the shRNA specifically for neuroligin-3 mRNA that was carried by AAVs and was microinjected in this cortical area (*Chang et al., 2006*; *Khatri et al., 2012*; *Pardridge, 2007*; *Pushparaj et al., 2008*; *Rao et al., 2009*). The microinjections of AAV-DJ/8-U6-mNlgn3-GFP (pAAV[shRNA]-U6-mNlgn3-EGFP) into the barrel cortex were done 3 days prior to the training paradigm of associative learning. This approach was expected to deteriorate the expression of neuroligin-3 in barrel cortical neurons, to prevent the formation of new synapse innervations from piriform, gustatory, and S1-Tr cortical neurons as well as to weaken the recruitment of associative memory neurons in the barrel cortex. The experiments in neuroligin-3 knockdown were conducted with AAV-mediated neural tracing and electrophysiological recording to examine its effectiveness on the morphology and function of associative memory cell recruitment. After the training paradigm of associative learning, those mice in the subgroups of shRNA and scramble-shRNA control within PSG were examined about their behaviors in memory tasks, convergent synapse innervations on barrel cortical neurons, as well as their electrophysiological activities in response to the whisker, olfactory, gustation, and tail stimulations. The quantities of barrel cortical neurons in response to those whisker, olfactory, gustatory, and tail signals were analyzed and compared in these two subgroups. The effectiveness of shRNA specific for neuroligin-3 on new synapse formation and associative memory cell recruitment was confirmed if the number of new synapse contacts and associative memory neurons

in the subgroup of neuroligin-3 shRNA was significantly lowered in comparison with shRNA control subgroup.

## Statistical analyses

All data are presented as arithmetic mean ± SEM. The statistical analyses of all of our data were conducted by using GraphPad Prism 9 publically. One-way ANOVA was used for the statistical comparisons of the changes in neuron activity and morphology between the groups of PSG and UPSG as well as the subgroups between *neuroligin-3* knockdown and scramble control. The $\chi^2$-test was used for the statistical comparison of changes in the percentage of recruited associative memory neurons identified by electrophysiological study among these groups. p-Values equally and above 0.05 in the comparison among groups were set to be no statistical differences. One asterisk, two asterisks, three asterisks, and four asterisks were presented to p<0.05, 0.01, 0.001, and 0.0001, respectively.

## Acknowledgements

This study is funded by the Natural Science Foundation of China (81971027, 81930033, and U2241209) to Jin-Hui Wang.

## Additional information

### Funding

| Funder | Grant reference number | Author |
| --- | --- | --- |
| National Natural Science Foundation of China | 81971027 | Jin-Hui Wang |
| National Natural Science Foundation of China | U2241209 | Jin-Hui Wang |
| National Natural Science Foundation of China | 81930033 | Jin-Hui Wang |

The funders had no role in study design, data collection and interpretation, or the decision to submit the work for publication.

### Author contributions

Yang Xu, Tian-liang Cui, Jia-yi Li, Bingchen Chen, Data curation, Formal analysis, Methodology; Jin-Hui Wang, Conceptualization, Resources, Supervision, Funding acquisition, Validation, Investigation, Writing - original draft, Project administration, Writing - review and editing, Experimental design

### Author ORCIDs

Jin-Hui Wang (iD) http://orcid.org/0000-0001-9452-1269

### Ethics

Experiments were conducted in accordance with guidelines and regulations by the Administration Office of Laboratory Animals in Beijing China. All experimental protocols were approved by the Institutional Animal Care and Use Committee in Administration Office of Laboratory Animals in Beijing China (B10831).

Reviewer #1 (Public Review): https://doi.org/10.7554/eLife.87969.3.sa1
Reviewer #2 (Public Review): https://doi.org/10.7554/eLife.87969.3.sa2
Author Response https://doi.org/10.7554/eLife.87969.3.sa3

## Additional files

### Supplementary files
• MDAR checklist

## Data availability

All data generated or analyzed during this study are included in the manuscript and supporting file.

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
