## [Editor Report · eLife assessment]

Multimodal experiences that for example contain both visual and tactile components are encoded as associative memories. This manuscript is a **valuable** contribution supporting structural and functional brain plasticity following associative training protocols that pair together different types of sensory stimuli. The results provide **solid** support for this plasticity being a basis for cross-modal associative memories.

---

## [Referee Report · Reviewer #1 (Public Review)]

This manuscript by Xu and colleagues addresses the important question of how multi-modal associations are encoded in the rodent brain. They use behavioral protocols to link stimuli to whisker movement and discover that the barrel cortex can be a hub for associations. Based on anatomical correlations, they suggest that structural plasticity between different areas can be linked to training. Moreover, they provide electrophysiological correlates that link to behavior and structure. Knock-down of nlg3 abolishes plasticity and learning.

This study provides an important contribution as to how multi-modal associations can be formed across cortical regions.

---

## [Referee Report · Reviewer #2 (Public Review)]

This manuscript by Xu et al. explores the potential joint storage/retrieval of associated signals in learning/memory and how that is encoded by some associative memory neurons using a mouse model. The authors examined mouse associative learning by pairing multimodal mouse learning including olfactory, tactile, gustatory, and pain/tail heating signals. The key finding is that after associative learning, barrel neurons respond to other multi-model stimulations. They found these barrel cortical neurons interconnect with other structures including piriform cortex, S1-Tr and gustatory cortical neurons. Further studies showed that Neuroligin 3 mediated the recruitment of associative memory neurons during paired stimulation group. The authors found that knockdown Neuroligin 3 in the barrel cortex suppressed the associative memory cell recruitment in the paired stimulation learning. Overall, this is an interesting study that reveals novel modalities associative learning involving multiple functionally connective cortical regions. Data presented are in general supporting their conclusions after revision.

---

## [Author Response]

The following is the authors’ response to the original reviews.

eLife assessment:Multimodal experiences that for example contain both visual and tactile components are encoded as associative memories. This manuscript is a valuable contribution supporting structural and functional brain plasticity following associative training protocols that pair together different types of sensory stimuli. The results provide solid support for this plasticity being a basis for cross-modal associative memories.

We appreciate eLife assessments to our discovery about the recruitment of associative memory neurons in cerebral cortices as a hub for the fulfillment of the first order and the second order of associative memory. Synapse interconnections among associative memory neurons mediate the reciprocal retrieval, the conversion and the translation of associated signals learnt in life span.

**Reviewer #1 (Public Review):**
This manuscript by Xu and colleagues addresses the important question of how multi-modal associations are encoded in the rodent brain. They use behavioral protocols to link stimuli to whisker movement and discover that the barrel cortex can be a hub for associations. Based on anatomical correlations, they suggest that structural plasticity between different areas can be linked to training. Moreover, they provide electrophysiological correlates that link to behavior and structure. Knock-down of nlg3 abolishes plasticity and learning. This study provides an important contribution as to how multi-modal associations can be formed across cortical regions.

We sincerely thank Reviewer one’s comments, which is a great driving force for us to move forward to reveal the specific roles of neural circuits in associative memory and its relevant cognitive activities and emotional reactions.

**Reviewer #2 (Public Review):**
This manuscript by Xu et al. explores the potential joint storage/retrieval of associated signals in learning/memory and how that is encoded by some associative memory neurons using a mouse model. The authors examined mouse associative learning by pairing multimodal mouse learning including olfactory, tactile, gustatory, and pain/tail heating signals. The key finding is that after associative learning, barrel neurons respond to other multi-model stimulations. They found these barrel cortical neurons interconnect with other structures including piriform cortex, S1-Tr and gustatory cortical neurons. Further studies showed that Neuroligin 3 mediated the recruitment of associative memory neurons during paired stimulation group. The authors found that knockdown Neuroligin 3 in the barrel cortex suppressed the associative memory cell recruitment in the paired stimulation learning. Overall, while the findings of this study are interesting, the concept of associative learning involving multiple functionally connective cortical regions is not that novel. While some data presented are convincing, the other seems to lack rigor. In addition, more details and clarification of the experimental methods are needed.Thank you so much for your comments on our studies in terms of the recruitment of associative memory neurons as the hub for the joint storage and reciprocal retrieval of multi-modal associated signals. You are right about that the concept of associative memory neuron and the new established interconnection among cerebral cortices for the formation of associative memory are not novel. The original finding has been reported by senior author’s lab many years ago, which has also been presented in a book by Jin-Hui Wang “Associative Memory Cells: Basic Units of Memory Trace” published by Springer-Nature 2019. In addition, we have made certain clarifications in our revision, but the detailed information about experimental approaches and concepts are expected to be seen in our previous publications and this book as well.
**Reviewer #1 (Recommendations For The Authors):**
I have two points that I find would strengthen the manuscript further:1. Associative memories are also based on specificity, which is not addressed in this manuscript. The authors could discuss this and also the magnitude of plasticity. In general, I would suggest also testing plasticity in response to a non-linked stimulus to prove specificity.

This a good point. In terms of the specificity of associative memory in our model, we have shown this point in our previous studies, such as Wang, et al. “Neurons in the barrel cortex turn into processing whisker and odor signals: a cellular mechanism for the storage and retrieval of associative signals”. Frontiers in Cellular Neuroscience 9-320:1-17 2015, and Jin-Hui Wang “Associative Memory Cells: Basic Units of Memory Trace” published by Springer-Nature 2019.

1. Nlg3 knock-down is a strong intervention. The authors could discuss the implications of interfering with synapse assembly and mechanistic implications at the synaptic level. It could help to compare the consequences of this intervention to a post-training lesion.

This is a good point. To prevent the possibility of post-training lesion by the intervention of Nlg3 knockdown, we have conducted the use of shRNA-scramble control. In addition, the discussion about the intervention of Nlg3 knockdown at synapse level has been added in our discussion.

1. In general, the clarity of the wording in some sections/sentences could be improved.

The rewording of certain sentences has been done in our revision.

**Reviewer #2 (Recommendations For The Authors):**
1. The writing of the manuscript needs major editing, there are grammatical errors even in the title. The extremely long introduction and discussion section with repeated details can be distracting from the main focus of the work.

This point has been taken during our revision.

1. Many bar graphs, such as Figure 5C and 5G, Figure 6C-6G, have low-resolution images, meaning that the axis titles and labels are unreadable.

The resolution of Figures have been improved in our revision.

1. The bar graph with data points and illustration in Figure 1E and 1G are misplaced.

This mistake has been corrected in our revision.

1. On page 23, Figure 2B, which layer(s) of the PC, S1Tr and GC were the images taken from? In the PSG group, why is there no red axon terminal signal observed in the three regions? does it indicate that there is no significant projection from the BC axon to PC, S1Tr, or GC neurons? Given that Thy1-YFP labeled glutamatergic neurons at PC, S1Tr, and GC and there is no discernable co-localization of yellow and green cells, can we assume that the glutamatergic neurons at PC, S1Tr, and GC are not involved in the associative learning after PSG paradigm? Lastly, the number of synapse contacts in Figure 2E is only 1-2 per 100um dendrite, but this is not quite consistent with the confocal images in Figure 2D. In Figure 2D, there are at least three tdTomato boutons on the cropped dendrite which is ~16um according to the scale bar.

If we magnify Figure 2B, we are able to see red boutons, which can be seen in Figure 2C with a higher magnification. In addition, the distribution of synapse contacts is variable, we have demonstrated the averaged values of synapse contacts over dendrites in Figure 2E, such that the single original image may not exactly same as the statistical data.

1. Figure 4C and Figure 8C, how were the percentages of associative neurons calculated after LFP recording? More details are needed on the method of this in vivo LFP/single unit recordings, including the spike sorting algorithm.

In the section of Results, the total number of neurons recorded in each of groups has been given. For instance, the neurons recorded from PSG mice (Figure 4) were 70, which was used as denominator. With the number of neurons that responded to two or more signals, the percentage of associative memory neurons recruited in associative learning was calculated. This information has been added in our revision (please see the section of Results).

1. The rationale for the authors choosing Neuroligin 3 as the target for investigating the formation of new synapse interconnections between BC, PC, S1Tr, and GC after PSG should be more clearly spelled out. Synaptic CAMs include SynCAM, NCAM, Neurexin, Cadherin et al all play a role in new synapse formation. Neuroligin 1 is expressed specifically in the CNS at excitatory synapses. Why did the authors choose to study Neuroligin 3 instead of Neuroligin 1?

This is a good point. Based on our previous data, miRNA-324 is upregulated during the associative learning by our mouse model, which degrades neuroligin-3 mRNA. The role of neuroligin-3 in the formation of new synapses and the recruitment of associative memory neurons is studied in this paper.

1. The behavioral results in Figure 5B-5G indicated that after pair-stimulation of WS-OS, WS-TS, or WS-GS, the memory learned in piriform, S1-Tr and gustatory cortical neurons can be retrieved from each other, by jumping over the barrel cortex. Is it possible that there is some direct interconnection formed between piriform, S1-Tr, and gustatory cortical neurons? Maybe they can try to do barrel cortical lesion or chemogenetic inhibition after PGS training and then repeat the behavioral tests as in Figure 5B-5G.

We have done experiments to examine the potential direct interconnection among piriform, S1-Tr and gustatory cortical neurons, after the associative learning about twelve days. We have no convincing data to support this possibility at this moment.

1. Some of the images showing the location of virus injections look VERY similar, such as Figure 3A left and right, Figures 7A and 7D. Larger variability of different animals/injection sites is definitely expected.

The injected viruses in Figure 3 and Figure 7 are different, since AAV-carried fluorescent proteins in different cortical areas are different. In addition, if we carefully enlarge the images in the right and left panels of Figure 3A, we will see that the areas of AAV transfection in morphology are different. The similarity of injection areas as Reviewer two claimed indicates the more precision of our virus-injection sites.

1. On page 49, are the green neurons in Figure 9B the BC cells? Just to be consistent, the authors should use the same color for BC cells as in Figure 9A. Also, label the primary and the secondary associative memory cells in Figure 9.

Figure 9 has been thoroughly changed in our revision.